# Direct observation of tunable thermal conductance at solid/porous crystalline solid interfaces induced by water adsorbates

Guang Wang [1,2], Hongzhao Fan [1,2], Jiawang Li[1], Zhigang Li[1] & Yanguang Zhou [1] ✉

Improving interfacial thermal transport is crucial for heat dissipation in devices with interfaces, such as electronics, buildings, and solar panels. Here, we design a strategy by utilizing the water adsorption-desorption process in porous metal-organic frameworks (MOFs) to tune the interfacial heat transfer, which could benefit their potential in cooling or heat dissipation applications. We observe a changeable thermal conductance across the solid/porous MOF interfaces owing to the dense water channel formed by the adsorbed water molecules in MOFs. Our experimental and/or modeling results show that the interfacial thermal conductance of $Au/Cu_3(BTC)_2$, $Au/Zr_6O_4(OH)_4(BDC)_6$ and Au/MOF-505 heterointerfaces is increased up to 7.1, 1.7 and 3.1 folds by this strategy, respectively, where $Cu_3(BTC)_2$ is referred to as HKUST-1 and $Zr_6O_4(OH)_4(BDC)_6$ is referred to as UiO-66. Our molecular dynamics simulations further show that the surface tension of Au layer will cause the adsorbed water molecules in MOFs to gather at the interfacial region. The dense water channel formed at the interfacial region can activate the high-frequency lattice vibrations and act as an additional thermal pathway, and then enhance heat transfer across the interfaces significantly. Our findings revealed the underlying mechanisms for tailoring thermal transport at the solid/porous MOF heterointerfaces by water adsorbates, which could motivate and benefit the new cooling system design based on MOFs.

The importance of cooling cannot be overstated, with its research extending from the hardware of the digital age (e.g., electronic cooling[1,2]) to the process of life (e.g., building cooling[3,4]). Recently, passive cooling using metal-organic frameworks (MOFs) has attracted considerable interest in the cooling of electronics[5], solar panels[6,7], and buildings[8] owing to its eco-friendly nature and zero-electricity characteristic. The corresponding cooling performance strongly depends on the thermal conductivity of MOFs and interfacial thermal conductance (ITC) between the objectives and MOFs. Unfortunately, MOFs typically possess a low thermal conductivity below 2 W/mK at room temperature[9–11] and are therefore regarded as poor thermal

conductors. Even worse, the adsorbed water molecules in MOFs may decrease the effective thermal conductivity further[12–14]. For example, Babaei et al. suggested that the thermal conductivity of MOF-199 (i.e., HKUST-1) can be reduced from 0.69 to 0.21 W/mK when water molecules are adsorbed[12,15]. Consequently, there is a small space to manipulate the intrinsic thermal transport properties of MOFs. Designing an effective interfacial heat dissipation channel across the objectives and MOFs may be the only feasible way to improve the corresponding cooling performance. This is critical for these cooling applications, where the saturated MOF component serves as the heat spreader. Therefore, the heat transfer inside the MOF and from the MOF to the

[1]Department of Mechanical and Aerospace Engineering, The Hong Kong University of Science and Technology, Clear Water Bay, Kowloon, Hong Kong SAR, China. [2]These authors contributed equally: Guang Wang, Hongzhao Fan. ✉e-mail: maeygzhou@ust.hk

ambient environment is dominated by the evaporation of the adsorbed water. The effect on the cooling performance caused by the low thermal conductivity of MOF and thermal resistance between MOF particles when compacting can be ignored. Till now, interfacial engineering using adhesion layer[16–18], nanostructures[19–24], chemical modification[25,26], and self-assembled monolayer (SAM)[27–31] has been widely applied to enhance the ITC. While it is possible to synthesize or fabricate the buffer layers, atomically controlling the structures of buffer layers is challenging and strongly limits its applications.

In this work, we design a sustainable and controllable strategy by utilizing a water adsorption process in porous MOFs to manipulate the interfacial heat transfer between Au and the MOF-199 (i.e., HKUST-1). Our frequency-domain thermoreflectance (FDTR)[32,33] measurements and/or molecular dynamics (MD) simulations show that the ITC of Au/HKUST-1, Au/UiO-66 and Au/MOF-505 heterointerfaces can be improved from 5.3, 12.5 and 6.9 MW/m$^2$K to 37.5, 22.9 and 21.4 MW/m$^2$K (~7.1, ~1.7 and ~3.1 times) via this strategy, respectively. The dense water channels formed by the adsorbed water molecules in HKUST-1 serve as additional thermal pathways and enhance the thermal energy across the interfaces significantly. The vibrational transmission coefficient function calculated by the frequency domain direct decomposition method (FDDDM)[34–36] further demonstrates that the thermal energy can be easily dissipated from Au to the MOF with adsorbates owing to the bridge effect of the adsorbed water molecules. Our findings provide new insights into thermal transport across MOFs (not only limited to HKUST-1) and their working objectives. We suggest a general strategy to introduce additional heat transfer channels between these MOFs and the objectives using adsorbed water, which

will greatly benefit the performance of MOFs-related cooling applications.

## Results and discussion
### Materials synthesis and characterization
Here, we choose a typical MOF (HKUST-1, i.e., MOF-199) which is widely used in gas storage and sensing[37], as a representative to investigate. The HKUST-1 crystals were synthesized by a hydrothermal method[13] (see the "Methods" section for details). Our synthesized HKUST-1 crystals showed an octahedral structure with a typical size of ~200 μm, and the triangle (111) facet[38] of the crystals was observed (Fig. 1a). The powder X-ray diffraction (PXRD) measurements indicated that the synthesized HKUST-1 crystals possessed a good crystallinity (Fig. 1b), which agrees with our calculated XRD spectrum. The HKUST-1 crystals were then mounted on a silicon wafer with a Kapton tape, and an Au transducer layer with a thickness of ~100 nm (see supplementary information (SI) Note 1 and Fig. S1 for details) was sputtered onto the crystals. We used the focused ion beam (FIB)-SEM system to get the cross-section of the Au/HKUST-1 heterointerfaces. The transmission electron microscopy (TEM) image (Fig. 1c) implied a clear and smooth interface between HKUST-1 and Au, where Au had good contact with the HKUST-1 crystal. The Au-coated HKUST-1 crystals were activated in a vacuum oven to evacuate the adsorbed water molecules during the synthesis process first and then immersed in deionized (DI) water for 40 mins. The Raman spectrum of the samples showed a redshift of the peak at 229–175 cm$^{-1}$ when the HKUST-1 crystal was soaked in water for 20 min (Fig. 1d), which was attributed to the water molecules' coordination with Cu–Cu units. There was no further change in the Raman

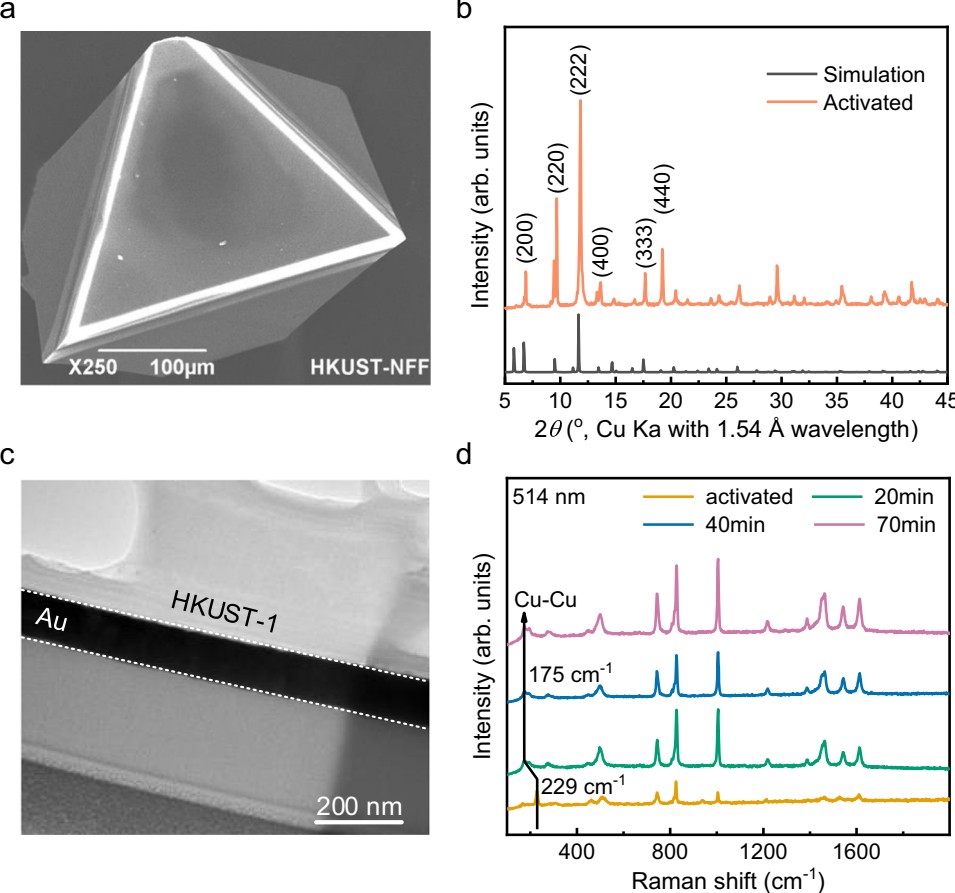

**Fig. 1 | Characterization of HKUST-1. a** The SEM image of HKUST-1 single crystal after activation. **b** The PXRD of HKUST-1 crystals after activation. **c** A cross-section TEM image of Au/HKUST-1 heterointerfaces prepared by FIB-SEM. **d** The Raman spectra of Au-coated HKUST-1 crystals after activation and immersed in water for 20, 40, and 70 min.

spectrum after prolonged soaking, which indicated that the HKUST-1 was fully saturated[39]. For activated samples, our FDTR measurements were conducted in a nitrogen environment to avoid water adsorption during the measurements. For samples with saturated water, the Au-coated HKUST-1 crystals were briefly dried by a compressed air flow to remove the remaining water film on the surface after being taken out from the DI water (see the "Methods" section for details).

## FDTR measurements

We then characterized the thermal transport properties of Au/HKUST-1 heterointerfaces using an optical pump-probe spectroscopy based on the FDTR. Here, Au also served as the transducer and was heated by the pump laser during the FDTR measurements. The schematic heat transfer model for the FDTR measurements is shown in the inset in Fig. 2. For activated samples where all the water molecules were released, the heat generated by the pump laser was transferred from the Au layer to the HKUST-1 framework directly. For samples with saturated water, the thermal energy was dissipated from the Au layer to both the HKUST-1 framework and the adsorbed water. The corresponding thermal conductivity was observed to be reduced compared to that of the activated HKUST-1, which agrees well with other experiments[20] and simulations[12,14] (see SI Fig. S5 for details).

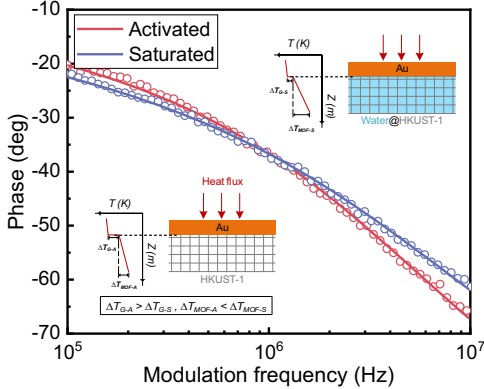

**Fig. 2 | Heat transport measurement with FDTR.** A representative FDTR signal phase as a function of the pump modulation frequency for the activated and fully saturated samples. The circle line is the raw data of FDTR measurements, and the solid line is the best-fitting line by the least-square method. The inset picture on the bottom left is a schematic and corresponding temperature profiles of activated Au/HKUST-1 heterointerfaces, and the top right inset picture is for the sample with saturated water molecules.

Then, high-quality Au/HKUST-1 heterointerfaces with flat and smooth surfaces (see SI Fig. S6 for details) were chosen in the FDTR measurements to ensure a good thermoreflectance response. Generally, the sample with parallel surfaces and uniform transducer layers can facilitate our measurements. This can be confirmed by the laser profiles obtained through the optical microscope (see S1.2 in SI Note1). Then, the phase lag between the pump beam and the probe beam measured by a lock-in amplifier in our FDTR measurements was fitted using a heat diffusion model[40,41]. The thermal conductivity ($\kappa$) of HKUST-1 and interfacial thermal conductance (ITC) between the Au layer and the HKUST-1 can be then determined. In the FDTR measurements, the intensity radius for the probe and the pump laser beams were 5 and 3.6 μm, respectively. The measured data were collected under a ×10 optical microscope. It is noted that $\kappa$ and ITC are two independent parameters in our thermal diffusion fitting model as the sensitivities of these two parameters are quite different (see SI Note 2 for details), and the fitting parameters used are summarized in Table S1. Figure 2 shows one representative FDTR measurement at room temperature. The measured thermal conductivity of the activated HKUST-1 crystals and HKUST-1 crystals with saturated water at the room temperature were 0.742 and 0.416 W/mK, respectively, which agreed well with other measured values[13]. The thermal conductance of Au/HKUST-1 heterointerfaces was found to increase from 5.17 to 31.5 MW/m²K when HKUST-1 adsorbed saturated water. It is noted that the water adsorbed in the saturated sample will not be evaporated according to the temperature rise estimation (see S3.1 in SI Note 3 for details).

In total, we have measured over one hundred Au/HKUST-1 heterointerface samples from six batches (see SI Note 3 for details), and the mean ITC is shown in Fig. 3a. For the activated samples, the measured ITC ranged from 3 to 8 MW/m² K. The average value was 5.33 ± 0.15 MW/m² K based on the Gaussian fitting of all measurements. When HKUST-1 adsorbed saturated water molecules, the mean ITC between the Au and the HKUST-1 increased to 21.66 ± 15.82 MW/m² K, which was four times higher than the activated samples. The maximum ITC was 37.9 MW/m² K for the sample with saturated water. Meanwhile, it is worth noting that the minimal ITC for the water-saturated sample is only 5.8 MW/m² K, which is close to the value of the activated samples. This may be attributed to the weak surface tension of the Au layer, which makes the water molecules in HKUST-1 not likely to be absorbed into the Au/HKUST-1 interfacial region. We emphasize that the Au layer in different Au/HKUST-1 heterointerfaces is different (see SI Fig. S7 for details), which results from a combination of factors such as impurities, surface roughness, and the crystal orientation of the Au film. Therefore, the surface tension of Au in some Au/HKUST-1 heterointerfaces may be too weak to gather the water molecules at the

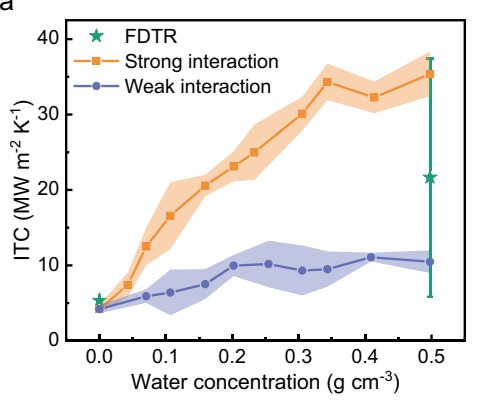

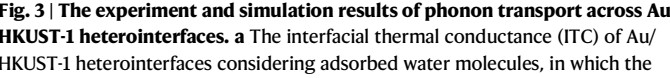

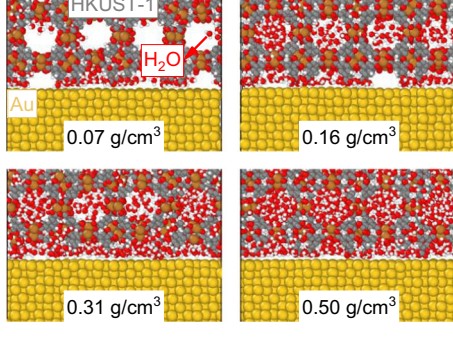

**Fig. 3 | The experiment and simulation results of phonon transport across Au/HKUST-1 heterointerfaces. a** The interfacial thermal conductance (ITC) of Au/HKUST-1 heterointerfaces considering adsorbed water molecules, in which the

shadow areas represent the stand errors of MD simulation results. **b** The atomistic Au/HKUST-1 heterointerfaces with various concentrations of adsorbed water molecules.

interfacial region (see our simulation analysis below for details). This is reasonable, as it is verified by the error difference between the activated and saturated samples. The relative error of the measured ITC of the saturated samples is much larger than that of the activated samples, which is caused by the water adsorption in HKUST-1 when all the other measurement conditions are the same. Nevertheless, the mean ITC is found to increase from 5.33 to 21.66 MW/m$^2$ K when the samples adsorb saturated water, which also implies that the adsorbed water molecules can largely affect the interfacial thermal transport between MOFs and substrates.

## Interfacial thermal conductance calculated using atomistic simulations

We also investigated the thermal transport across the Au/HKUST-1 heterointerfaces via nonequilibrium molecular dynamics (NEMD) simulations (see the "Methods" section and SI Note 4 for calculation details). We first calculated the ITC of Au/HKUST-1 heterointerfaces without absorbed water molecules. The calculated ITC was $4.20 \pm 0.57$ MW/m$^2$ K, which was close to our measured value of $5.33 \pm 0.15$ MW/m$^2$ K (Fig. 3a). The agreement of the ITC between experiments and simulations indicated that the interatomic interactions between the Au and the HKUST-1 were properly described in our MD simulations. We then studied the thermal transport across the Au/HKUST-1 heterointerfaces considering adsorbed water molecules. As discussed above, the structures of the Au layers in our Au/HKUST-1 heterointerfaces are different. It is known that the interaction between Au and water molecules strongly depends on the structure of Au[42,43]. Therefore, two sets of interatomic potentials are fitted to depict the strong and weak Au/HKUST-1 heterointerfaces by comparing our calculated ITC with the measured ITC. The detailed fitting procedure for the interatomic interactions can be found in SI Note 4.

For the models with strong Au–water molecule interactions, the ITC increased first and then saturated to ~35 MW/m$^2$ K with the adsorbed water molecules. The ITC for the Au/HKUST-1 heterointerfaces with saturated water molecules agreed well with the upper value of our experimental measurements. For the models with weak Au–water molecule interactions, the ITC increased to a stable value of ~10 MW/m$^2$ K when HKUST-1 absorbed saturated water molecules, which was close to the lower value of the experimental results. We next investigated the dynamic water adsorption process in Au/HKUST-1 heterointerface models. Water molecules were initially adsorbed in HKUST-1 randomly (the water adsorption process can be found in SI Note 4). During the relaxation period, the adsorbed water molecules were found to cluster in the cages of HKUST-1 or be adsorbed in the interface region (Fig. 3b). The former behavior was because of the long-range electrostatic interactions among water molecules and their thermal motion, which may affect the thermal transport properties of HKUST-1 considering absorbed water molecules[14]. The latter resulted from the intrinsic interactions between water molecules and Au, which may largely increase the ITC of Au/HKUST-1 heterointerfaces.

## Underlying mechanisms

We further calculated the transmission coefficient function of Au/HKUST-1 heterointerfaces with/without absorbed water molecules using FDDDM[34–36]. The transmission coefficient function quantitatively characterizes the thermal energy exchange capability of vibrations. Our results showed that the vibrational transmission coefficient was generally increased when the water molecules were absorbed into the systems (Fig. 4a and b). The transmission coefficient function for the heterointerfaces with strong Au–water molecule interactions was found to be larger than that with weak Au–water molecule interactions. This was because more water molecules were adsorbed into the interfacial region for the heterointerfaces with strong Au–water molecule interactions.

Meanwhile, vibrations with frequencies higher than 4 THz were found to contribute little to the ITC for the pristine Au/HKUST-1 heterointerfaces. When a large number of water molecules were adsorbed into the interfacial region (e.g., the density of adsorbed water molecules was 0.5 g/cm$^3$), some vibrations with frequencies higher than 4 THz could also transport thermal energy across the heterointerfaces (Fig. 4a and b). Although vibrations with frequencies higher than 4 THz can exist on both sides of the Au/HKUST-1 interface, the contribution from vibrations higher than 4 THz to interfacial thermal transport could only be activated by enough adsorbed water molecules. This is because of the bridge effect resulting from the adsorbed water molecules in the interfacial region. To analyze the bridge effect, we calculated the vibrational density of states (VDOSs) of Au, HKUST-1, water molecules, and HKUST-1 with water molecules in the region near the interface (Fig. 4c). While there is an overlap below 4 THz between the VDOS of Au and that of HKUST-1, their large mismatch implies that the heat transport from Au to HKUST-1 is inefficient. Consequently, the ITC of the pristine Au/HKUST-1 heterointerfaces is low. However, the overlap of the VDOS between Au and water molecules is broad and apparent (the inset of Fig. 4c), which leads to an increased ITC when water molecules are adsorbed into the interface region (Fig. 3).

Before closing, we quantitatively characterized the thermal energy across heterointerfaces from the Au layer to water molecules and the HKUST-1 framework. Figure 4d shows the ITC contributions resulting from the two channels of Au/water molecules and the Au/HKUST-1 framework. The density of the adsorbed water molecules was 0, 0.2, and 0.5 g/cm$^3$ in our Au/HKUST-1 heterointerfaces. We found that the ITC resulting from the Au/HKUST-1 framework channel was independent of the density of the adsorbed water molecules and almost a constant value of ~5 MW/m$^2$ K. The corresponding ITC spectrum also showed a weak dependence on the density of the adsorbed water molecules (SI Note 4). However, both the total ITC and ITC contributed by the channel of Au/water molecules increased significantly with the adsorbed water molecules (Fig. 4d and SI Note 4). Our results reveal that the adsorbed water molecules do not hinder the thermal transport between Au and HKUST-1 and provide extra thermal pathways to dissipate heat from Au.

## The generalizability of the proposed strategy

To show the generalizability of our proposed strategy, we further investigated the thermal transport in two more MOF/substrate systems (i.e., UiO-66/Au and MOF-505/Au heterointerfaces) using either experiments or MD simulations. The chosen UiO-66[44,45] and MOF-505[46] here show good water adsorption capacity and stability. Our FDTR measurements show that the ITC between UiO-66 crystals and the Au film increases from $16.4 \pm 2.9$ to $19.7 \pm 3.15$ MW/m$^2$ K when saturated water is adsorbed (see S5.1 in SI Note 5 for details). Our MD simulations also show that the ITC of activated Au/MOF-505 heterointerface is $7.72 \pm 0.82$ MW/m$^2$ K, and increases to $19.56 \pm 1.81$ MW/m$^2$ K when saturated water molecules are adsorbed (see S5.2 in SI Note 5 for details).

In summary, we have designed a tunable strategy utilizing the adsorbed water in porous MOFs to manipulate the thermal transport across Au/MOF interfaces. Our FDTR measurements and/or simulations showed that a maximum ITC ~37.9, ~22.9, and ~21.4 MW/m$^2$ K could be achieved when saturated water molecules were adsorbed in HKUST-1, UiO-66, and MOF-505, respectively. These values were ~7.1, ~1.7, and ~3.1 times higher than the ITCs of the activated Au/HKUST-1, Au/UiO-66, and Au/MOF-505 heterointerfaces. Our NEMD simulations further demonstrated that this ITC enhancement was because of the bridge effect of the dense water channel at the Au/MOF interfacial region formed by adsorbed water molecules in MOFs. The adsorbed water molecules at the interfacial region could not only activate the contributions of high-frequency lattice vibrations but also act as an additional thermal pathway. As a result, the thermal energy can be

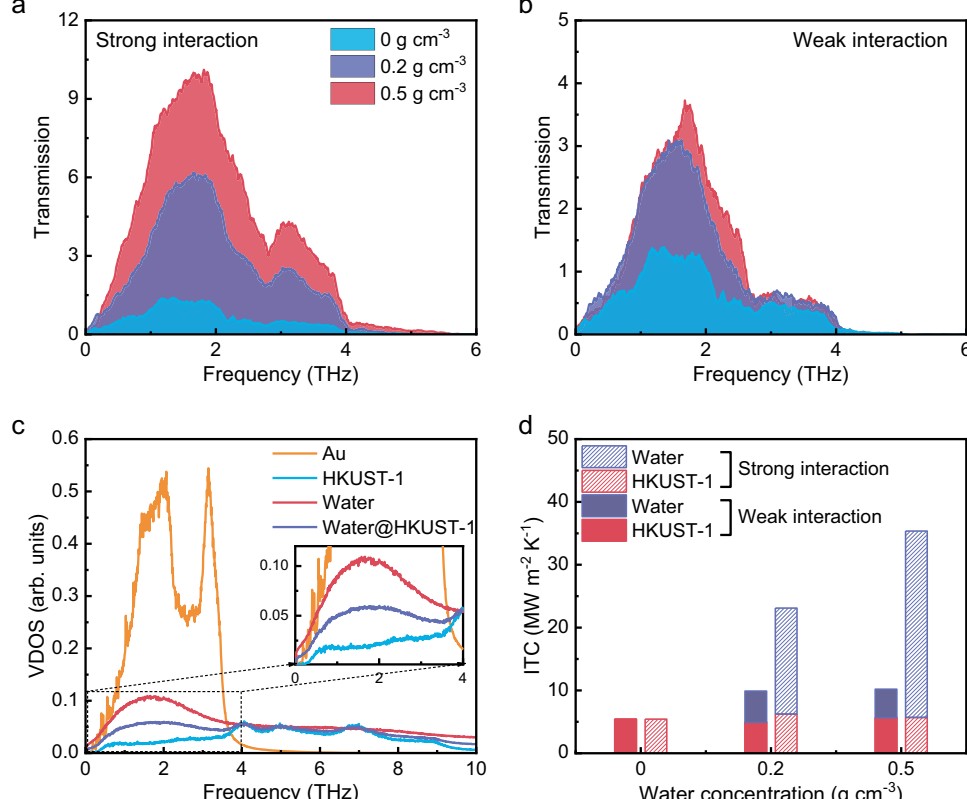

**Fig. 4 | The calculated thermal transport properties across Au/HKUST-1 heterointerfaces. a** The transmission coefficient function for the heterointerfaces with strong Au-water molecule interaction and different water concentrations. **b** The transmission coefficient function for the heterointerfaces with weak Au–water molecule interaction and different water concentrations. **c** The VDOSs of Au, HKUST-1, water molecules and HKUST-1 with water molecules in the region near the strong-interaction interface with a water density of 0.5 g/cm³. **d** The interfacial thermal conductance (ITC) resulted from the two channels of Au/water molecules and Au/HKUST-1 framework. The concentration of adsorbed water molecules in HKUST-1 is 0, 0.2, and 0.5 g/cm³.

easily dissipated from Au to the MOFs with adsorbates owing to the bridge effect of the dense water channel, which is further confirmed by our calculated vibrational transmission coefficient function. Our work here proposed a new strategy based on the water adsorption–desorption process in porous MOFs to manipulate the heat transfer across MOFs/solid interfaces. The underlying mechanism of the heat transfer across the MOFs/solid interfaces provided here will guide the design of effective cooling or heat dissipation systems using MOFs.

## Methods
### Synthesis and characterization of Au/HKUST-1 heterointerfaces
The HKUST-1 crystals were synthesized by a hydrothermal method[13]. First, 5 g Benzene-1,3,5-tricarboxylic acid (Bidepharm, 98%) and 0.6 g oxalic acid dihydrate (Aladdin, 99.5%) were dissolved in a mixture of 100 mL EtOH (VWR, 96%) and 10 mL DMF (RCI Labscan Limited, 99.8%). The solution with 16.5 g Cu(NO$_3$)$_2$·3H$_2$O (Aladdin, 99%) and 90 mL DI water was slowly added to the above linker solution. The resulting suspension was stirred for 1 h to ensure its thorough mixing, and then sealed in a capped jar. The suspension was then put in an 80 °C pre-heated oven for 48 h. A mixture of blue HKUST-1 crystals and the insoluble white precipitate was then formed, and the precipitate was removed by adding fresh ethanol and pipetting out the white suspension. This process might be repeated several times until pure HKUST-1 crystals are obtained. Finally, these HKUST-1 crystals were rinsed in ethanol for 12 h and activated in a 150 °C vacuum oven for 20 h.

The HKUST-1 crystals were then mounted on a silicon wafer with a Kapton tape, and a ~100 nm Au layer was sputtered on the crystals to form Au/HKUST-1 heterointerface samples (Discovery 18, Denton Vacuum). The surface morphology of Au/HKUST-1 heterointerfaces

was characterized by scanning electron microscopy (SEM, JSM-6490 Jeol). The PXRD of HKUST-1 was also measured at room temperature by powder diffractometer X'pert Pro (PANalytical, CuKα1 radiation, $\lambda = 1.54056$ Å). A cross-section of Au/HKUST-1 heterointerfaces was prepared by the FIB-SEM dual beam system (FEI Helio ns G4 UX) using a standard lift-out procedure with a final milling step. The thickness of the specimen was ~100 nm, and the bright field images of the specimen's cross-section were captured by scanning transmission electron microscopy (STEM, JEM-ARM200F JEOL). The Raman spectroscopy measurements were performed using a Raman spectrometer (InVia, Renishaw) with an excitation wavelength of 514 nm. For the activated samples, the measurements were conducted in a vacuum chamber to avoid the adsorption of moisture during the test.

### FDTR measurements
To avoid moisture adsorption during our FDTR measurements for the activated sample, the Au/HKUST-1 heterointerface specimen was mounted in a semi-sealed chamber (Instec) with a slow nitrogen flow. For each FDTR test, the 1/e² diameter of the pump laser was measured using the beam offset method and the laser spot was fitted by the Gaussian profiles[47]. Typically, the radius of the pump and probe laser were around 3.6 and 5 μm, respectively. The phase lag between the probe laser and pump laser was collected by the lock-in amplifier (HF2LI, Zurich), and fitted by the heat diffusion model to obtain the thermal properties. The FDTR lasers were swept across the samples' surface under the optical microscope to find the smooth and flat area for good thermoreflectance. Each sample spot was swept five times to reduce the noise in experiments. Based on the sensitivity analysis, the thermal conductivity and the ITC between Au and HKUST-1 crystal

could be determined at the same time (see SI Note 2 for details). To measure the thermal properties of Au/HKUST-1 heterointerfaces with adsorbed water, the activated samples were immersed in DI water for 40 min. The immersion time in our experiments was long enough for the HKUST-1 crystals to absorb saturated water, which was confirmed by the Raman measurements (Fig. 1d) and other references[13,39]. Then, we applied the slow compression air flow to remove the water film on the Au/HKUST-1 samples, which were taken out from the water. The thermal properties of Au/HKUST-1 samples with saturated water were measured in the room environment using FDTR.

## MD simulation

In this paper, the NEMD simulations were performed to investigate the thermal transport of the Au/HKUST-1 heterointerfaces considering water adsorption. All MD simulations were implemented by the Large-scale Atomic/Molecular Massively Parallel Simulator (LAMMPS) software[48]. The size of the simulation systems was 5.2 nm × 5.2 nm × 41.2 nm. To calculate the interfacial thermal conductance, a symmetrical model was used here. Au atoms were located on two sides of the system, and the HKUST-1 frameworks with/without adsorbed water molecules were in the middle region (see SI Note 4 for details). The embedded atom method potential was used to describe the interactions among Au atoms. A forcefield developed based on first-principles calculations was applied to describe the interaction of the HKUST-1 framework[49]. The interactions among water molecules were depicted by the extended simple point charge model (SPC/E)[50]. Parameters extracted from the universal force field (UFF) were adopted to describe the interactions between Au and HKUST-1[51]. The interactions between Au and water molecules were fitted based on parameters from references[52] and our experimental measurements. The long-range electrostatic interactions were considered in our simulations and solved by the particle–particle–particle mesh method[53].

During the MD simulations, the systems were first relaxed in an isothermal-isobaric ensemble and then a canonical ensemble to release the residual stress. Following, the NEMD simulations were implemented to calculate interfacial thermal conductance. Periodic boundary conditions were applied along both $x$ and $y$ directions. The fixed boundary condition was applied along the $z$ direction. The temperature gradient was then generated by Langevin thermostats. The heat sink and source were placed at two sides of the systems near the fixed atoms. The temperature of the heat source and sink was set as 350 and 250 K, respectively. When the steady-temperature gradient was built, the accumulative thermal energies $\Delta E$ added or subtracted to the system by thermostats were recorded for 2.5 ns. The heat current $Q$ across the interface was then calculated by linearly fitting the slope of $\Delta E$ versus simulation time. The temperature difference $\Delta T$ was obtained by linearly extrapolating the temperature distributions at two sides of the interface. The interfacial thermal conductance is ITC = $Q/(\Delta T \cdot A)$, where $A$ is the cross-sectional area of the systems.

## FDDDM calculations

The interfacial spectral thermal conductance and the corresponding transmission coefficient function are calculated by the FDDDM method[34–36], which is in the framework of NEMD simulations. During NEMD simulations, the heat current transferred across the interface can be calculated by

$$Q_{\text{left} \to \text{right}} = \sum_{i \in \text{left}} \sum_{j \in \text{right}} \left\langle \frac{\partial U_j}{\partial \mathbf{r}_i} \cdot \mathbf{v_i} - \frac{\partial U_i}{\partial \mathbf{r}_i} \cdot \mathbf{v_j} \right\rangle \quad (1)$$

where $U$ represents the potential energy, $\mathbf{v_i}$ is atomic velocity and $\mathbf{r_i}$ is atomic position. The atomic velocity and position are recorded during the NEMD simulation. Then, the spectral heat current can be obtained via

$$Q(\omega) = \text{Re} \sum_{i \in \text{left}} \sum_{j \in \text{right}} \int_{-\infty}^{+\infty} \left\langle \frac{\partial U_j}{\partial \mathbf{r_i}} \Big|_\tau \cdot \mathbf{v_i}(0) - \frac{\partial U_i}{\partial \mathbf{r_j}} \Big|_\tau \cdot \mathbf{v_j}(0) \right\rangle e^{i\omega\tau} d\tau \quad (2)$$

Since the potential used to depict the interface is a two-body interaction, the spectral heat current can be simplified into

$$Q(\omega) = 2\text{Re} \sum_{i \in \text{left}} \sum_{j \in \text{right}} \int_{-\infty}^{+\infty} \mathbf{F_{ij}} \cdot \mathbf{v_i}(0) e^{i\omega\tau} d\tau \quad (3)$$

where the $\mathbf{F_{ij}}$ is the interatomic force from atom $j$ exerted on atom $i$. During the NEMD simulation processes, the atomic trajectories, forces, and velocities of atoms in the interfacial region with a thickness of ~2.5 nm during the sampled 500 ps were used as inputs for our FDDDM calculations. To minimize the uncertainty caused by sampling in MD simulations, the transmission functions were independently calculated three times and then taken the ensemble average.

Once the frequency-dependent heat current is obtained, the phonon transmission function can be then estimated based on the Landauer theory[54–56]. In the Landauer theory, the heat current spectrum from the left lead to the right lead through a junction connecting two leads at two different equilibrium heat-bath temperatures (i.e., $T_L$ and $T_R$) is written in the form of

$$Q(\omega) = \hbar\omega[n_L(\omega) - n_R(\omega)]\Gamma(\omega) \quad (4)$$

where $n$ is the equilibrium phonon distribution function at heat-bath temperatures and has the classical limit form of $n_{L \text{ or } R}(\omega) = k_B T_{L \text{ or } R}/\hbar\omega$ in molecular dynamics simulations, in which $\Gamma(\omega)$ is the phonon transmission function. The spectral thermal conductance $G(\omega)$ is then written as

$$G(\omega) = \frac{\hbar\omega[n_L(\omega) - n_R(\omega)] \cdot \Gamma(\omega)}{A(T_L - T_R)} = \frac{\hbar\omega\Delta n(\omega) \cdot \Gamma(\omega)}{A\Delta T}$$
$$\approx \frac{\hbar\omega\partial n(\omega) \cdot \Gamma(\omega)}{A\partial T} \overset{\text{classical limit}}{\approx} \frac{k_B\Gamma(\omega)}{A} \quad (5)$$

It should be noted that Eq. (5) is only valid when the temperature gradient is kept in the linear regime. Therefore, the phonon transmission function in NEMD simulations can be calculated using $\Gamma(\omega) = Q(\omega)/k_B\Delta T$.

## Reporting summary

Further information on research design is available in the Nature Portfolio Reporting Summary linked to this article.

# Data availability

Data underpinning the figures that support this work are available within the paper and its Supplementary Information files.

# Code availability

The code that supports the findings of this study is available from the corresponding author upon request.

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

## Acknowledgements

Y.Z. thanks the Equipment Competition fund (REC20EGR14) and the Sustainable and Smart Campus as a living lab fund (FS105) from the Hong Kong University of Science and Technology (HKUST), the open fund from the State Key Laboratory of Clean Energy Utilization (ZJU-CEU2022009) and the ASPIRE Seed Fund (ASPIRE2022#1) from the ASPIRE League. Z.L and Y.Z. acknowledge the fund from the Research Grants Council of the Hong Kong Special Administrative Region under Grant C6020-22G. Y.Z. also thanks for the Hong Kong SciTech Pioneers Award from the Y-LOT Foundation. The authors are grateful to the Materials Characterization and Preparation Facility (MCPF) of HKUST for their assistance in experimental characterizations.

## Author contributions

Y.Z. conceived the idea; Z.L. and Y.Z. supervised the project; G.W. designed the experiments and conducted the material synthesis, characterization, and performance investigation; H.F. did the calculations; J.L. conducted the characterization; G.W., F.H. and Y.Z. prepared the manuscript.; G.W., H.F., Z.L. and Y.Z. reviewed, and revised the manuscript.

## Competing interests

The authors declare no competing interests.
