## [Peer Review File · Nature Communications]

Direct Observation of Tunable Thermal Conductance at Solid/
porous Crystalline Solid Interfaces Induced by Water
AdsorbentsREVIEWER COMMENTS

Reviewer #1 (Remarks to the Author):

Porous MOFs are subject to their own thermal conductivity and ITC. In response to the low thermal conductivity of MOFs, the author proposed a sustainable and controllable strategy that utilizes water adsorbed in porous MOFs to regulate the ITC of Au /HKUST-1. The influence of the density of the adsorbed water molecules on ITC was discussed through FDTR experiments and MD simulations. In terms of experiments, various verification methods are used to analyze the change process of the interface. In MD simulations, the changes of ITC under different interface conditions are discussed. The results show that the water molecules gathered at the interface play a bridging role and the thermal energy can be easily dissipated from Au to MOFs. The research still has certain significance. Therefore, I recommend a minor revision of this manuscript. Below I outline the problems that need to be addressed.

1. In sample preparations, the Au transducer is sputtered on the MOFs and a reference glass simultaneously to calibrate the thickness of Au layer. The MOF crystals are quite small and taped on a substrate. How to make sure the surfaces of the MOFs are parallel to the substrates? Will this affect the thickness of the Au transducer layer, since this should be a relatively sensitive parameter in FDTR data processing?
2. The FDTR in this work utilizes an ultrafast laser as the light source. What are the specifications of the laser, such as the pulse width and the energy of each pulse? Will the instantaneous heating of pump pulses affect the adsorbed water in the materials?
3. The vibrational mode near the interface may be different from the atoms far away from the interface. The authors should make further discussion about the VDOS spatial variation.
4. The authors mentioned "The dense water channel at the interfacial region could not only activate the high frequency lattice vibration, but also act as an additional thermal path." in lines 193-194. From Figure 4 (c), it can be observed that the overlapping area between HKUST-1 and water becomes wider, indicating that water reduces the vibration mismatch between Au /HKUST-1. However, the cutoff frequencies of the materials are not presented. The authors should give more evidences and explanations to support their conclusion.
5. In Figure 4 (b), in the 3-4 THz range, water with an adsorption density of 0.2 g/cm³ seems to have a greater contribution to ITC than water with a higher adsorption density. What is the reason for this?

Reviewer #2 (Remarks to the Author):

Thermal conductivity is a critical parameter for MOFs in many practical applications. However, experimental measurement of the thermal conductivity of MOF crystals is of great challenge due to its small size. Wang et al. reported a method for measuring thermal conductivity of MOF crystals. The authors attempted to enhance interfacial heat transfer through water between MOFs and Au films, however, in practical use, these waters do not exist stably. In fact, compared with the interfacial thermal resistance, the contact thermal resistance between MOF particles is larger and more critical. Given this work is submitted to Nat. Commun., I suggest that deep consideration should be taken for whether this manuscript deserves an opportunity of revision. In addition, I would remind the author to consider the following aspects.

1. The reliability of the method to determine the thickness of Au transducer is doubtful. By measuring the thickness of the Au film sputtered on the glass, the authors equivalent the

thickness of the Au film on the MOFs. Due to the different wettability of glass and MOFs, the way of Au film formation on the surface of both may be different. In addition, it is possible to obtain a film of uniform thickness on glass, but it is difficult to obtain a film of uniform thickness on the surface of MOFs due to the octahedral structure of HKUST-1 microcrystalline.

2. In terms of test results, the measured ITC ranged from 5.8 to 37.9 MW/m²K, and the measurements were highly volatile. The authors attribute this to the effects of impurities, surface roughness and the crystal orientation of the Au film, which are unavoidable in practice.

3. HKUST-1 is a class of MOFs with large grain sizes, and the effectiveness of this method for MOFs with small grain sizes requires further consideration.

Response Letter

Dear Reviewers,

Thank you for reviewing our manuscript entitled "Direct Observation of Tunable Thermal Conductance at Solid/porous Crystalline Solid Interfaces Induced by Water Adsorbents" (Manuscript ID: NCOMMS-23-10826). We appreciate your evaluation and believe that your comments have improved our revised manuscript substantially. In the following, we address all review comments (in cool blue for clarity) in detail and provide responses as well as the resulting revisions (in strong blue) to our manuscript. A copy of the manuscript with revisions highlighted in yellow is also provided for reference.

Response to the first referee

Comment 0

Porous MOFs are subject to their own thermal conductivity and ITC. In response to the low thermal conductivity of MOFs, the author proposed a sustainable and controllable strategy that utilizes water adsorbed in porous MOFs to regulate the ITC of Au /HKUST-1. The influence of the density of the adsorbed water molecules on ITC was discussed through FDTR experiments and MD simulations. In terms of experiments, various verification methods are used to analyze the change process of the interface. In MD simulations, the changes of ITC under different interface conditions are discussed. The results show that the water molecules gathered at the interface play a bridging role and the thermal energy can be easily dissipated from Au to MOFs. The research

still has certain significance. Therefore, I recommend a minor revision of this manuscript. Below I outline the problems that need to be addressed.

Our response:

We thank the reviewer for the positive evaluation and constructive comments on our manuscript, which greatly helped us improve the quality of the manuscript. We have addressed all the comments point by point in the revised manuscript.

Comment 1

In sample preparations, the Au transducer is sputtered on the MOFs and a reference glass simultaneously to calibrate the thickness of Au layer. The MOF crystals are quite small and taped on a substrate. How to make sure the surfaces of the MOFs are parallel to the substrates? Will this affect the thickness of the Au transducer layer, since this should be a relatively sensitive parameter in FDTR data processing?

Our response:

We thank the reviewer for this comment.

During sample preparation process, many HKUST-1 crystals were attached to a Si substrate. Au film was then deposited on the Si substrate with HKUST-1 crystals and a reference glass slice which was partly covered with a PI tape. To ensure that the thickness of Au on the HKUST-1 crystal and glass slice are the same, we used the focused ion beam-scanning microscope electron (FIB-SEM) to get a cross-section of the Au/HKUST-1 (**Figure 1c**). The TEM analysis software DigitalMicrography is used to measure the thickness of the Au film of the Au/HKUST-1 sample (**Figure R1**). The thicknesses of Au film at three different locations are 102.95 nm, 104.88 nm,

and 105.17 nm, respectively. The mean thickness of this sample is regarded as ~ 104.3 nm, which is similar to the thickness (i.e., ~ 106 nm) of Au film on the reference glass slice as obtained by AFM (**Figure S1** in the supplementary materials). Therefore, we can think they have the same thickness with acceptable error. Besides, we also include an uncertainty of $\sim 5\%$ of the thickness in the FDTR fitting to consider the error caused by the slight variation of Au thickness. In fact, a uniform, flat, and smooth Au transducer film is critical for our following FDTR measurements to obtain a clear thermal reflectance signal. Furthermore, we need to find suitable regions for our FDTR measurements.

Figure R1 (added as **Figure S2**). The TEM image of a cross-section of a typical Au/HKUST-1 sample which is prepared by FIB-SEM.

It is true that the size of HKUST-1 crystals in our samples is small and ranges from several tens to several hundreds of μm . Meanwhile, the surface of HKUST-1 crystals on the substrate is different. In our study, we can only measure the thermal transport properties of samples with HKUST-1 crystals parallel to the substrate. Here, the pump laser profile is used to confirm that the surface of HKUST-1 crystals is parallel to the substrate. The laser profile under the optical microscope differs for the HKUST-1 samples with various surfaces. We use a beam offset method to obtain the laser

profile. For instance, the laser profile on an HKUST-1 sample with a surface parallel to the substrate (**Figure R2a**) is circular (**Figure R2b**), whereas on the HKUST-1 sample with a surface not parallel to the substrate (**Figure R2d**) has a conical shape (**Figure R2e**). Furthermore, because of the octahedral structure of HKUST-1 crystals, the HKUST-1 surface needs to be parallel to the Si substrate to achieve a uniform sputtered Au film. The laser profile on HKUST-1 parallel surfaces is fitted using the Gaussian function, and the derived laser spot radius is $\sim 3.6 \mu\text{m}$ (**Figure R2c**), whereas the laser profile on HKUST-1 tilted surfaces cannot be fitted using the Gaussian function (**Figure R2f**). To ensure the accuracy of our measurements, we measure these HKUST-1 samples with relatively large grain sizes (i.e., several hundreds of μm), as shown in **Figure 1a**. Therefore, the grain size of our chosen HKUST-1 samples is believed to be large enough for sputtering a thin Au film. We also note that the thickness of the sputtered Au film is around 105 nm, which is much smaller than the HKUST-1 grain size, enabling us to find an appropriate region for the FDTR measurements.

Figure R2 (added as **Figure S3**). The laser profiles on the samples with parallel and tilted surfaces with respect to the Si substrate. a, The sample with a parallel surface to the Si substrate and b, its corresponding laser spot under the microscope. c, The Gaussian beam profile on the samples with parallel surfaces to the Si substrate. d, the sample with a tilted surface to the Si substrate and e, its corresponding laser spot under the microscope. f, The Gaussian beam profile on the samples with tilted surfaces to the Si substrate.

Based on the reviewer's comment, we have updated the corresponding contents in the main text and supplementary information:

- **(Page 2 in our revised supplementary information)**: To make sure the thickness on the reference glass slice is the same as the Au layer on HKUST-1 crystals, we get a cross-section of this sample using the focused ion beam-scanning microscope electron (FIB-SEM). The TEM analysis software DigitalMicrography is used to measure the thickness of the Au film of the Au/HKUST-1 sample (**Figure S2**). The thicknesses of Au film at three different locations are 102.95 nm, 104.88 nm, and 105.17 nm, respectively. The mean thickness of this sample is regarded as ~104.3nm, which is similar to the thickness (i.e., ~106 nm) of Au film on the reference glass slice as obtained by AFM. Therefore, we can think they have the same thickness with acceptable error. Besides, we also include an uncertainty of ~5% of the thickness in the FDTR fitting to consider the error caused by the slight variation of Au thickness.
- **(Add one more subsection S1.2 in our revised supplementary information)**: As the size of HKUST-1 crystals in our samples is small and ranges from several tens to several hundreds of μm . Meanwhile, the surface of HKUST-1 crystals on the substrate is different. In our study, we can only measure the thermal transport properties of samples with HKUST-1 crystals parallel to the substrate. Here, the pump laser profile is used to confirm

that the surface of HKUST-1 crystals is parallel to the substrate. The laser profile under the optical microscope differs for the HKUST-1 samples with various surfaces. We use a beam offset method to obtain the laser profile. For instance, the laser profile on an HKUST-1 sample with a surface parallel (**Figure S3a**) to the substrate is circular (**Figure S3b**), whereas on the HKUST-1 sample with a surface not parallel to the substrate (**Figure S3d**) has a conical shape (**Figure S3e**). Furthermore, because of the octahedral structure of HKUST-1 crystals, the HKUST-1 surface needs to be parallel to the Si substrate to achieve a uniform sputtered Au film. The laser profile on HKUST-1 parallel surfaces is fitted using the Gaussian function, and the derived laser spot radius is $\sim 3.6 \mu\text{m}$ (**Figure S3c**), whereas the laser profile on HKUST-1 tilted surfaces cannot be fitted using the Gaussian function (**Figure S3f**). To ensure the accuracy of our measurements, we measure these HKUST-1 samples with relatively large grain sizes (i.e., several hundreds of μm), as shown in **Figure 1a**. Therefore, the grain size of our chosen HKUST-1 samples is believed to be large enough for sputtering a thin Au film, and is much larger than the transducer thickness, which enables us to find an appropriate region for the FDTR measurements.

- (**Page 5 in our revised manuscript**): Generally, the sample with parallel surfaces and uniform transducer layers can facilitate our measurements. This can be confirmed by the laser profiles obtained through the optical microscope (**see S1.2 in SI Note1**).

Comment 2

The FDTR in this work utilizes an ultrafast laser as the light source. What are the specifications of the laser, such as the pulse width and the energy of each pulse? Will the instantaneous heating of pump pulses affect the absorbed water in the materials?

Our response:

We thank the reviewer for this very good comment.

In our FDTR apparatus, one pump laser acts as a heat source, and the other probe laser detects the corresponding temperature change through the surface reflectivity change of the Au transducer. Two continuous-wave (cw) lasers are used in our system. The pump uses a 365 mW (nominal power) diode laser with a wavelength of 445 nm (OBIS 445-365C) which can be modulated by the lock-in amplifier. The probe is a 20 mW diode laser with a wavelength of 532 nm (OBIS LS 532-20).

It is true that the water molecules in the interface region will be evaporated when the power of the heat source is too high. To minimize the temperature rise caused by the heat source, we use a filter to reduce the power of the applied heat source to 10% of the original value. The heating power on the sample surface is around 2.7 mW after a 27 mW original laser pump source goes through the filter. This heating power can generate a clear thermorefectance signal in our FDTR apparatus but will not evaporate the water, as described in the following.

The temperature rise in FDTR measurements can be estimated based on the radially symmetric heat diffusion equation [Rev. Sci. Instrum., **75**, 5119 (2004)]. For a semi-infinite solid, the frequency domain solution for a surface heated by a unit power at angular frequency ω is

$$g(r) = \frac{\exp(-qr)}{2\pi\kappa r} \quad (1)$$

and

$$q^2 = \frac{i\omega}{D} \quad (2)$$

where κ is the thermal conductivity of the solid, D is the thermal diffusivity, and r is the radial coordinate. We can then apply the Hankel transform [J. Heat Transfer., **121**, 954 (1999)] to $g(r)$ for the convolution of $g(r)$ with laser intensities in a co-aligned pump and probe system. The Hankel transform of $g(r)$ is

$$G(k) = 2\pi \int_0^{\infty} g(r) J_0(2\pi kr) r dr = \frac{1}{\kappa \sqrt{(4\pi^2 k^2 + q^2)^{1/2}}} \quad (3)$$

where the J_0 is zeroth-order Bessel function of the first kind. If the surface is heated by a pump laser beam with a Gaussian distribution intensity $p(r)$, which has a form of

$$p(r) = \frac{2A}{\pi w_0^2} \exp\left(\frac{-2r^2}{w_0^2}\right) \quad (4)$$

in which, A is the amplitude of the adsorbed heat power, w_0 is the $1/e^2$ radius of the pump. The laser intensity in the frequency domain can be obtained using the Hankel transform

$$P(k) = A \exp\left(\frac{-\pi^2 k^2 w_0^2}{2}\right) \quad (5)$$

The temperature distribution at the surface is therefore expressed as the inverse transform of the product of $G(k)$ and $P(k)$

$$\theta(r) = 2\pi \int_0^{\infty} P(k) G(k) J_0(2\pi kr) k dk \quad (6)$$

In our FDTR apparatus, the probe laser detects the temperature rise based on the change of surface reflectivity. Considering the probe laser intensity also has a Gaussian distribution, the weighted average temperature distribution acquired by a probe beam at frequency w_1 is

$$\Delta T = \frac{4}{w_1^2} \int_0^{\infty} \theta(r) \exp\left(\frac{-2r^2}{w_1^2}\right) r dr \quad (7)$$

We use the Matlab script developed by Braun *et al.* [J. Heat Transfer. **140**, 052801 (2018)] based on Eq. (7) to calculate the temperature rise of our systems caused by the applied heat source. The temperature rise results from two parts: 1, a steady-state response resulting from the average adsorbed power, and 2, a modulated response from oscillations at the modulation frequency about the average power. It is widely accepted that the steady-state temperature rise dominates the global temperature rise [J. Appl. Phys. **121**, 175107 (2017)].

In our calculations, a two-layer model is used to represent our Au/HKUST-1 systems. The Au layer has a thickness of 105 nm, a heat capacity of 2.56 MJ/m³K, and an isotropic thermal conductivity of 183 W/mK. The saturated HKUST-1 layer is assumed to be a semi-infinite layer, which has a heat capacity of 3.618 MJ/m³K and an isotropic thermal conductivity of 0.416 W/mK. The interfacial thermal conductance (ITC) between the Au layer and the saturated HKUST-1 layer is set as 21.66 MW/m²K (i.e., the mean value of our experimental results). Our results show that the largest temperature rise of the laser spot is ~9 K and ~1.0 K when the modulation frequency is 2000 Hz (**Figure R3a**) and 10 MHz (**Figure R3b**), respectively. In our calculations, we have considered that only 20% of the 532 nm wavelength laser is absorbed by the Au transducer [Rev. Sci. Instrum. **84**, 104904 (2013)]. The temperature rise for the low modulation frequency is higher than that for the high modulation frequency, which agrees with other references [Rev. Sci. Instrum., **75**, 5119, (2004)]. As suggested by Jiang *et al.* [J. Phys. D: Appl. Phys. **54**, 035304 (2021)] that the temperature rise caused by the applied heat source should not exceed 10 K or 10% of the absolute temperature. Therefore, the temperature rise in our FDTR measurements is believed to be acceptable. Meanwhile, it is known that the thermal conductivity of the Au transducer is much higher than that of HKUST-1, which makes the temperature rise caused by the applied heat source in MOFs much smaller than the above-calculated values.

Our FDTR measurements are conducted at ~ 298 K, and the maximum temperature in the samples can be regarded as ~ 307 K, considering the temperature rise caused by the applied heat source. Our thermal gravimetric analysis (TGA) shows that the weight loss ratio of the saturated HKUST-1 at 307 K is 4.6% (**Figure R4a**). The modulation frequency of the pump laser in FDTR measurements ranges from 2000 Hz to 50 MHz, which indicates that the maximum temperature rise caused by the applied heat source should be much lower than 9 K. The maximum weight loss ratio of our samples caused by the applied heat source in all our FDTR measurements should be much smaller than 4.6%. As a result, the applied heat source has little influence on our measured thermal properties. Furthermore, we swept 5 times at the same location using the same pump laser and found that the measured phase lag curves were almost the same (**Figure R4b**). This again indicates that the applied heat source has little influence on our measured results.

Figure R3 (added as Figure S9). The temperature rise as a function of radius on the Au transducer layer at modulation frequencies of a, 2000 Hz and b, 10 MHz.

Figure R4 (added as Figure S10). a, The TGA test of saturated HKUST-1 with a ramp rate of 10K/min. b, The phase lag of one typical saturated sample which was swept 5 times at the same position.

Based on the reviewer's comment, we have added one subsection S 3.1 in Supplementary Note 3 in the revised supplementary material.

- **(Supplementary Note 3 in supplementary material):**

S 3.1 The temperature rise effect on the water adsorbed in HKUST-1

The FDTR uses the pump laser as the heat source, and the temperature rise can be detected by the probe laser. Whereas, if the temperature is too high, the water collected by the HKUST-1 might be evaporated, which may affect the measurements. To minimize the temperature rise caused by the heat source, we use a filter to reduce the power of the applied heat source to 10% of the original value. The heating power on the sample surface is around 2.7 mW after a 27 mW original laser pump source goes through the filter. This heating power can generate a clear thermorefectance signal in our FDTR apparatus but will not evaporate the water, as described in the following.

The temperature rise in FDTR measurements can be estimated based on the radially symmetric heat diffusion equation³. For a semi-infinite solid, the frequency domain solution for a surface heated by a unit power at angular frequency ω is

$$g(r) = \frac{\exp(-qr)}{2\pi\kappa r} \quad (8)$$

and

$$q^2 = \frac{i\omega}{D} \quad (9)$$

where κ is the thermal conductivity of the solid, D is the thermal diffusivity, and r is the radial coordinate. We can then apply the Hankel transform⁴ to $g(r)$ for the convolution of $g(r)$ with laser intensities in a co-aligned pump and probe system. The Hankel transform of $g(r)$ is

$$G(k) = 2\pi \int_0^{\infty} g(r) J_0(2\pi kr) r dr = \frac{1}{\kappa \sqrt{(4\pi^2 k^2 + q^2)^{1/2}}} \quad (10)$$

where the J_0 is the zeroth-order Bessel function of the first kind. If the surface is heated by a pump laser beam with a Gaussian distribution intensity $p(r)$, which has a form of

$$p(r) = \frac{2A}{\pi w_0^2} \exp\left(\frac{-2r^2}{w_0^2}\right) \quad (11)$$

in which, A is the amplitude of the adsorbed heat power, w_0 is the $1/e^2$ radius of the pump.

The laser intensity in the frequency domain can be obtained using the Hankel transform

$$P(k) = A \exp\left(\frac{-\pi^2 k^2 w_0^2}{2}\right) \quad (12)$$

The temperature distribution at the surface is therefore expressed as the inverse transform of the product of $G(k)$ and $P(k)$

$$\theta(r) = 2\pi \int_0^{\infty} P(k)G(k)J_0(2\pi kr)kdk \quad (13)$$

In our FDTR apparatus, the probe laser detects the temperature rise based on the change of surface reflectivity. Considering the probe laser intensity also has a Gaussian distribution, the weighted average temperature distribution acquired by a probe beam at frequency ω_1 is

$$\Delta T = \frac{4}{w_1^2} \int_0^{\infty} \theta(r) \exp\left(\frac{-2r^2}{w_1^2}\right) r dr \quad (14)$$

We use the Matlab script developed by Braun et al.⁵ based on Eq. (7) to calculate the temperature rise of our systems caused by the applied heat source. The temperature rise results from two parts: 1, a steady-state response resulting from the average adsorbed power, and 2, a modulated response from oscillations at the modulation frequency about the average power. It is widely accepted that the steady-state temperature rise dominates the global temperature rise⁶.

In our calculations, a two-layer model is used to represent our Au/HKUST-1 systems. The Au layer has a thickness of 105 nm, a heat capacity of 2.56 MJ/m³K, and an isotropic thermal conductivity of 183 W/mK. The saturated HKUST-1 layer is assumed to be a semi-infinite layer, which has a heat capacity of 3.618 MJ/m³K and an isotropic thermal conductivity of 0.416 W/mK. The interfacial thermal conductance (ITC) between the Au layer and the saturated HKUST-1 layer is set as 21.66 MW/m²K (i.e., the mean value of our experimental results). Our results show that the largest temperature rise of the laser spot is ~9 K and ~1.0 K when the modulation frequency is 2000 Hz (Figure S9a) and 10 MHz (Figure S9b), respectively. In our calculations, we have considered that only 20% of the a 532 nm wavelength laser is absorbed by the Au transducer⁷. The temperature rise for the

low modulation frequency is higher than that for the high modulation frequency, which agrees with other references³. As suggested by Jiang et al.⁸ that the temperature rise caused by the applied heat source should not exceed 10 K or 10% of the absolute temperature. Therefore, the temperature rise in our FDTR measurements is believed to be acceptable. Meanwhile, it is known that the thermal conductivity of the Au transducer is much higher than that of HKUST-1, which makes the temperature rise caused by the applied heat source in MOFs much smaller than the above-calculated values.

Our FDTR measurements are conducted at ~298 K, and the maximum temperature in the samples can be regarded as ~307 K, considering the temperature rise caused by the applied heat source. Our thermal gravimetric analysis (TGA) shows that the weight loss ratio of the saturated HKUST-1 at 307 K is 4.6% (Figure S10a). The modulation frequency of the pump laser in FDTR measurements ranges from 2000 Hz to 50 MHz, which indicates that the maximum temperature rise caused by the applied heat source should be much lower than 9 K. The maximum weight loss ratio of our samples caused by the applied heat source in all our FDTR measurements should be much smaller than 4.6%. As a result, the applied heat source has little influence on our measured thermal properties. Furthermore, we swept 5 times at the same location using the same pump laser and found that the measured phase lag curves were almost the same (Figure S10b). This once again indicates that the applied heat source has little influence on our measured results.

- **(Page 5 in our revised manuscript):** It is noted that the water adsorbed in the saturated sample will not be evaporated according to the temperature rise estimation (see S3.1 in SI Note 3 for details).

[3] Cahill, D. G. *Analysis of heat flow in layered structures for time-domain thermoreflectance*. *Rev. Sci. Instrum.* **75**, 5119–5122 (2004).

[4] Ohsone, Y., Wu, G., Dryden, J., Zok, F. & Majumdar, A. *Optical Measurement of Thermal Contact Conductance Between Wafer-Like Thin Solid Samples*. *J. Heat Transf.* **121**, 954–963 (1999).

[5] Braun, J. L., Szejewski, C. J., Giri, A. & Hopkins, P. E. *On the Steady-State Temperature Rise During Laser Heating of Multilayer Thin Films in Optical Pump–Probe Techniques*. *J. Heat Transf.* **140**, (2018).

[6] Braun, J. L. & Hopkins, P. E. *Upper limit to the thermal penetration depth during modulated heating of multilayer thin films with pulsed and continuous wave lasers: A numerical study*. *J. Appl. Phys.* **121**, 175107 (2017).

[7] Yang, J., Maragliano, C. & Schmidt, A. J. *Thermal property microscopy with frequency domain thermoreflectance*. *Rev. Sci. Instrum.* **84**, 104904 (2013).

Comment 3

The vibrational mode near the interface may be different from the atoms far away from the interface. The authors should make further discussion about the VDOS spatial variation?

Our response:

We thank the reviewer for this good comment.

It is true that the vibrations near the interface may be different from that in the area far away from the interface. This is because the atoms in the interfacial region are strongly affected by both of the two contacting materials. For example, both experiments [Nat. Commun., **12**, 6901 (2021)] and simulations [Sci. Rep., **6**, 23139 (2016)] showed that the phonon modes at the Si/Ge interfaces and the VDOS of Si at the interfacial region are different from that far away from the interface.

Based on the energy distribution along the direction perpendicular to the interface (**Figure R5a**), the interfacial region can be regarded to have a thickness of ~0.6 nm, which includes three layers of Au atoms. We then calculate the VDOS of Au at various positions with different distances from the interface. Our results show that the VDOS of the first layer of Au near the interface has been

changed due to the interface effect, and the VDOS of Au will converge with the distance from the interface (**Figure R5**). We further find that the HKUST with or without water molecules has little influence on the VDOS of Au atoms near the interface (**Figure R5**). This is because the interaction among Au atoms is stronger than that between Au and water molecules or HKUST-1. Furthermore, the thickness of the Au layers affected by the interface is ~ 0.6 nm, which is much smaller than the dimension of our Au/HKUST-1 samples. Therefore, it should be reasonable to assume that the interfaces in our FDTR fitting models and NEMD calculations have a zero thickness.

Figure R5 (added as Figure S15). a, The potential energy distribution of Au atoms near the interface, in which each point means the averaged potential energy of one layer Au atoms. b, The VDOS of Au atoms in different positions at bare Au interface. c, The VDOS of Au atoms in different positions at Au-HKUST-1 interface with water density of 0 g/cm^3 . d, The VDOS of Au atoms in different positions at Au-HKUST-1 interface with water density of 0.5 g/cm^3 .

Based on the reviewer's comment, we have added more explanations and a subsection S 4.4 in Supplementary Note 4:

- **(Supplementary Note 4 in supplementary material):**

S 4.4 The vibrational density of state (VDOS) near the interface

For the interfacial thermal transport, the vibrations near the interface may be different from those in the area far away from the interface. This is because the atoms in the interfacial region are strongly affected by both of the two contacting materials. To analyze the impacts from interfacial interaction to atomic vibrations, we defined the interface region and discussed the VDOS spatial variation. Based on the energy distribution along the direction perpendicular to the interface (**Figure S15**), the interfacial region can be regarded to have a thickness of ~ 0.6 nm, which includes three layers of Au atoms. We then calculate the VDOS of Au at various positions with different distances from the interface. Our results show that the VDOS of the first layer of Au near the interface has been changed due to the interface effect, and the VDOS of Au will converge with the distance from the interface (**Figure S15 b-d**). We further find that the HKUST with or without water molecules has little influence on the VDOS of Au atoms near the interface (**Figure S15 c-d**). This is because the interaction among Au atoms is stronger than that between Au and water molecules or HKUST-1. Furthermore, the thickness of the Au layers affected by the interface is ~ 0.6 nm, which is much smaller than the dimension of our Au/HKUST-1 samples. Therefore, it should be reasonable to assume that the interfaces in our FDTR fitting models and NEMD calculations have a zero thickness.

Comment 4

The authors mentioned "The dense water channel at the interfacial region could not only activate the high frequency lattice vibration, but also act as an additional thermal path." in lines 193-194. From Figure 4 (c), it can be observed that the overlapping area between HKUST-1 and water becomes wider, indicating that water reduces the vibration mismatch between Au /HKUST-1. However, the cutoff frequencies of the materials are not presented. The authors should give more evidences and explanations to support their conclusion.

Our response:

We thank the reviewer for this very good comment.

The vibrational cutoff frequency of the Au substrate and HKUST-1 is ~ 5 THz and ~ 40 THz (**Figure 4c**), respectively. Here, the hydrogen atoms are not considered as their corresponding vibrations are in the frequency range of ~ 95 THz, which are not occupied at room temperature. Our calculated transmission function shows that only these vibrations with frequencies smaller than 5 THz contribute to the interfacial thermal transport across Au/HKUST-1 interfaces (**Figures 4a** and **4b**). We also find that only these vibrations with frequencies smaller than 4 THz transfer thermal energy across the Au/activated HKUST-1 interfaces, even if the vibrations with frequencies ranging from 4 to 5 THz can exist on both sides of our Au/HKUST-1 samples (**Figures 4a** and **4b**). These vibrations with frequencies of 4~5 THz can transport thermal energy across the interfaces when enough water molecules are adsorbed in the interfacial region (**Figure 4a**). We therefore believe that the vibrational thermal transport channels in the frequencies of 4~5 THz result from the dense water molecules in the interfacial region. Our calculated VDOS of Au, HKUST-1, and water molecules further show that the adsorbed water molecules may act as a

bridge to reduce the vibrational mismatch between Au and HKUST-1 (**Figure 4c**). Meanwhile, we emphasize that the ITC across the water-saturated Au/ HKUST-1 interfaces carried by these vibrations is small compared to the total ITC (i.e., ~1.8%).

Based on the reviewer's comment, we have added some discussions in our revised manuscript.

- **(Page 8 in our revised manuscript):** Although the vibrations with frequencies higher than 4 THz can exist on both sides of Au/HKUST-1 interface, the contribution from vibrations higher than 4 THz to interfacial thermal transport could only be activated by enough adsorbed water molecules.
- **(Page 8 in our revised manuscript):** To analyze the bridge effect, we calculated the vibrational density of states (VDOSs) of Au, HKUST-1, water molecules and HKUST-1 with water molecules in the region near the interface (**Figure 4c**).
- **(Page 9 in our revised manuscript):** The adsorbed water molecules at the interfacial region could not only activate the contributions of high-frequency lattice vibrations but also act as an additional thermal pathway.

Comment 5

In Figure 4 (b), in the 3-4 THz range, water with an adsorption density of 0.2 g/cm³ seems to have a greater contribution to ITC than water with a higher adsorption density. What is the reason for this?

Our response:

We thank the reviewer for this very good comment.

In our analysis, we apply the FDDDM to calculate the vibrational interfacial transmission function. The atomic trajectories, forces, and velocities of atoms in the interfacial region with a thickness of ~ 2.5 nm during the sampled 500 ps were used as inputs for our FDDDM calculations. It is known that the adsorbed water molecules can diffuse in HKUST-1 and the interfacial region during the MD simulations. The systems of the Au/HKUST-1 with water molecules at various times should be different. Therefore, the ITC and interfacial transmission function should be calculated with the ensemble average. As shown in **Figure 3a**, all our calculated ITC are averaged using five independent MD simulations. However, the vibrational interfacial transmission function was calculated using one MD simulation. This may be the reason for the vibrations in the frequency range of 3-4 THz for the systems with a water adsorption density of 0.2 g/cm^3 have a greater contribution to ITC.

Based on the reviewer's comment, we have calculated the vibrational transmission function of each system using three independent MD simulations and updated the corresponding results (**Figure 4**) in our revised manuscript. We have also added some explanations of the calculation details in our revised manuscript.

- **(Page 13 in our revised manuscript):** During the NEMD simulation processes, the atomic trajectories, forces, and velocities of atoms in the interfacial region with a thickness of ~ 2.5 nm during the sampled 500 ps were used as inputs for our FDDDM calculations. To minish the uncertainty caused by sampling in MD simulations, the transmission functions were independently calculated three times and then do the the ensemble average.

Response to the second referee

Comment 0

Thermal conductivity is a critical parameter for MOFs in many practical applications. However, experimental measurement of the thermal conductivity of MOF crystals is of great challenge due to its small size. Wang et al. reported a method for measuring thermal conductivity of MOF crystals. The authors attempted to enhance interfacial heat transfer through water between MOFs and Au films, however, in practical use, these waters do not exist stably. In fact, compared with the interfacial thermal resistance, the contact thermal resistance between MOF particles is larger and more critical. Given this work is submitted to Nat. Commun., I suggest that deep consideration should be taken for whether this manuscript deserves an opportunity of revision. In addition, I would remind the author to consider the following aspects.

Our response:

We appreciate the reviewer for dedicating the time to thoroughly review our manuscript and providing constructive comments and suggestions. It is true that in some MOF-related applications, water doesn't exist stably. For example, when the MOF is used for CO₂ and N₂ separation, the vapor in the gas should be removed before separation if the MOF is not stable to water [Chem. Rev., **112**,724 (2012)]. Therefore, for those systems that don't allow water to exist, our proposed strategy in this manuscript is invalid. However, in many thermal-related applications (e.g., cooling or heat pump) based on the adsorption and desorption process of MOFs, the corresponding thermal transport properties in the MOF-based systems are critical. For instance, our previous study has applied a MOF-801 based cooling coating to cool the solar panels [Droplet **2**, e32 (2023)]. In that application, we paint the MOF-801 based cooling coating on the backside of the solar panels and

utilize the atmospheric water adsorption-desorption process in the coating to cool the temperature of the solar panels. Our results show that the temperature of solar panels can be reduced by over 10 °C at most using such a strategy. Wang *et al.* [Joule **4**, 435 (2021)] also utilize the atmospheric water adsorption and desorption process of MIL-101(Cr) for chips' cooling and find that the cooling power can be as high as 0.5 W. It is obvious that the thermal transport across the MOF with or without atmospheric water/substrate (i.e., solar panel or electronics) is critical for the corresponding cooling performance in these applications mentioned above.

Meanwhile, we agree that the thermal resistance of the MOF-based system should include three parts: the intrinsic thermal resistance of MOFs and the substrate, the interfacial thermal resistance between the substrate and MOF, and the interfacial thermal resistance between MOF particles. It is true that there exist many grain boundaries between MOF particles as MOF particles are generally tightly compacted in real applications (e.g., the above-mentioned cooling paint). These interfaces between MOF particles, of course, will affect the corresponding performance in real applications. However, it is challenging to quantitatively characterize the interfacial thermal resistance between MOF particles as the MOF particles are sparsely dispersed on the substrate in our experiments.

Furthermore, our study here mainly focuses on the thermal transport across the interface between MOF and the substrate, in which the substrate and MOF are treated as heat source and a heat spreader, respectively. As shown in **Figure R6**, the demonstration in our study can be simplified as a three-layer model, where the interface between Au and HKUST-1 is a layer without thickness. Therefore, the heat transfer from the heat source to the ambient air can be divided into three steps: 1) The heat is generated on the Au surface and conducted within the Au substrate; 2) The heat conductively transfers across the interface from the Au substrate to the HKUST-1; 3) The heat

adsorbed in HKUST-1 will dissipate the ambient air mainly through the evaporation of the adsorbed water owing to its high enthalpy. In step 3, the heat dissipated through the conduction heat transfer is small as the thermal conductivity of HKUST-1 with and without water is relatively low [Droplet 2, e32 (2023)]. Indeed, the thermal resistance between MOF particles will strongly limit the conduction heat transfer in HKUST-1. However, the heat in HKUST-1 is mainly dissipated through the evaporation of the adsorbed water. Therefore, the conductive thermal resistance of the HKUST-1 side has little influence on the applications mentioned above. In contrast, the ITC between the substrate and HKUST-1 will strongly affect the performance of the mentioned applications. For the same temperature difference between the Au substrate and HKUST-1, the thermal energy transferred from the substrate to HKUST-1 can be increased around 4 times during a specific time when the ITC is improved from 5.33 MW/m²K (i.e., the ITC of the activated Au/HKUST-1 samples) to 21.66 MW/m²K (i.e., the mean ITC of the saturated Au/HKUST-1 samples). Therefore, in these applications mentioned above (e.g., evaporative cooling using MOFs), the ITC between the substrate and MOFs is one of the critical factors that determines the corresponding performance.

Figure R6. Illustration of the heat dissipation in an Au/HKUST-1 system with saturated water, where Q_{input} is the input heat flux. $Q_{c-Au@ITC}$ is the heat flux transferred in Au transducer and across the interface by heat conduction, which will result in a temperature difference ΔT_c between the upper surfaces of Au and HKUST-1. t_{Au} and κ_{Au} denote the thickness and thermal conductivity of Au, respectively. ITC represents the interfacial thermal conductance. The output heat flux Q_{output} is equal to input heat flux Q_{input} , and can be divided into two parts: the heat energy transferred by HKUST-1 conduction (Q_{c-MOF}) and the evaporation of adsorbed water ($Q_{Evaporation}$), where the latter dominates.

Based on the reviewer's comment, we have revised the related contents:

- **(Page 2 in the revised manuscript):** This is critical for these cooling applications, where the saturated MOF component serves as the heat spreader. Therefore, the heat transfer inside the MOF and from the MOF to the ambient environment is dominated by the evaporation of the adsorbed water. The effect on the cooling performance caused by the low thermal conductivity of MOF and thermal resistance between MOF particles when compacting can be ignored.

Comment 1

The reliability of the method to determine the thickness of Au transducer is doubtful. By measuring the thickness of the Au film sputtered on the glass, the authors equivalent the thickness of the Au film on the MOFs. Due to the different wettability of glass and MOFs, the way of Au film formation on the surface of both may be different. In addition, it is possible to obtain a film of uniform thickness on glass, but it is difficult to obtain a film of uniform thickness on the surface of MOFs due to the octahedral structure of HKUST-1 microcrystalline?

Our response:

We thank the reviewer for this very good comment.

It is true that the thickness of the Au film deposited on the glass slice may be a little different from the Au film on the HKUST-1 crystals as the wettability and morphology of both systems are different. To ensure the thickness of Au on the HKUST-1 crystal and glass slice is the same, we use the FIB-SEM to get a cross-section of the Au/HKUST-1 (**Figure 1c**). The TEM analysis software DigitalMicrography is used to measure the thickness of the Au film of the Au/HKUST-1 sample (**Figure R1**). The thicknesses of Au film at three different locations are 102.95 nm, 104.88 nm, and 105.17 nm, respectively. The mean thickness of this sample is ~ 104.3 nm, which is similar to the thickness (i.e., ~ 106 nm) of Au film on the reference glass slice as obtained by AFM (**Figure S1** in the supplementary materials). Therefore, we can think they have the same thickness with acceptable error. Besides, we also include an uncertainty $\sim 5\%$ of the thickness in the FDTR fitting to consider the error caused by the small variation of Au thickness. In fact, a uniform, flat, and smooth Au transducer film is critical for our following FDTR measurements to obtain a clear thermal reflectance signal. As the HKUST-1 crystals have octahedral structures, the exposed surface may not be parallel to the substrate, which results in a non-uniform and unevenly deposited Au film. We therefore need to find suitable regions in which the exposed surface of HKUST-1 crystals is parallel to the Au substrate for our FDTR measurements.

During the sample preparation process, many HKUST-1 crystals were attached to a Si substrate. The Au film was then deposited on the Si substrate with HKUST-1 crystals. Under the optical microscope, we can distinguish whether the surfaces of the samples are parallel to the Au substrate or not based on the corresponding laser profile. For instance, the laser profile on an HKUST-1

sample with a surface parallel to the substrate is circular (**Figure R2b**), whereas on the HKUST-1 sample with a surface not parallel to the substrate has a conical shape (**Figure R2e**). Furthermore, because of the octahedral structure of HKUST-1 crystals, the HKUST-1 surface needs to be parallel to the Si substrate to achieve a uniform sputtered Au film. The laser profile on HKUST-1 parallel surfaces is fitted using the Gaussian function (**Figure R2c**), whereas the laser profile on HKUST-1 tilted surfaces cannot be fitted using the Gaussian function (**Figure R2f**). The fitted Gaussian function based on the laser profile is critical for our following FDTR measurements.

The size of HKUST-1 crystals in our samples is ranging several tens to several hundreds of μm . To ensure the accuracy of our measurements, we measure these HKUST-1 samples with relatively large grain sizes (i.e., several hundreds of μm), as shown in **Figure 1a**. The grain size of our chosen HKUST-1 samples is therefore believed to be large enough for sputtering a thin Au film. We also note that the thickness of the sputtered Au film is around 105 nm, which is much smaller than the HKUST-1 grain size. It is therefore easy for us to find suitable regions for the FDTR measurements.

Based on the reviewer's comment, we have updated the corresponding contents in the main text and supplementary information:

- (**Page 2 in our revised supplementary information**): To make sure the thickness on the reference glass slice is the same as the Au layer on HKUST-1 crystals, we get a cross-section of this sample using the focused ion beam-scanning microscope electron (FIB-SEM). The TEM analysis software DigitalMicrography is used to measure the thickness of the Au film of the Au/HKUST-1 sample (**Figure S2**). The thicknesses of Au film at three different locations are 102.95 nm, 104.88 nm, and 105.17 nm, respectively. The mean thickness of this sample is regarded as $\sim 104.3\text{nm}$, which is similar to the thickness (i.e.,

~106 nm) of Au film on the reference glass slice as obtained by AFM. Therefore, we can think they have the same thickness with acceptable error. Besides, we also include an uncertainty of ~5% of the thickness in the FDTR fitting to consider the error caused by the slight variation of Au thickness.

- **(Add one more subsection S1.2 in our revised supplementary information):** As the size of HKUST-1 crystals in our samples is small and ranges from several tens to several hundreds of μm . Meanwhile, the surface of HKUST-1 crystals on the substrate is different. In our study, we can only measure the thermal transport properties of samples with HKUST-1 crystals parallel to the substrate. Here, the pump laser profile is used to confirm that the surface of HKUST-1 crystals is parallel to the substrate. The laser profile under the optical microscope differs for the HKUST-1 samples with various surfaces. We use a beam offset method to obtain the laser profile. For instance, the laser profile on an HKUST-1 sample with a surface parallel (**Figure S3a**) to the substrate is circular (**Figure S3b**), whereas on the HKUST-1 sample with a surface not parallel to the substrate (**Figure S3d**) has a conical shape (**Figure S3e**). Furthermore, because of the octahedral structure of HKUST-1 crystals, the HKUST-1 surface needs to be parallel to the Si substrate to achieve a uniform sputtered Au film. The laser profile on HKUST-1 parallel surfaces is fitted using the Gaussian function, and the derived laser spot radius is $\sim 3.6 \mu\text{m}$ (**Figure S3c**), whereas the laser profile on HKUST-1 tilted surfaces cannot be fitted using the Gaussian function (**Figure S3f**). To ensure the accuracy of our measurements, we measure these HKUST-1 samples with relatively large grain sizes (i.e., several hundreds of μm), as shown in **Figure 1a**. Therefore, the grain size of our chosen HKUST-1 samples is believed to be large

enough for sputtering a thin Au film, and is much larger than the transducer thickness, which enables us to find an appropriate region for the FDTR measurements.

- (Page 5 in our revised manuscript): Generally, the sample with parallel surfaces and uniform transducer layers can facilitate our measurements. This can be confirmed by the laser profiles obtained through the optical microscope (see S1.2 in SI Note1).

Comment 2

In terms of test results, the measured ITC ranged from 5.8 to 37.9 MW/m²K, and the measurements were highly volatile. The authors attribute this to the effects of impurities, surface roughness and the crystal orientation of the Au film, which are unavoidable in practice.

Our response:

We thank the reviewer for this good comment.

We emphasize that the samples in our experiments should have different surface roughness, impurities, and crystal orientation of the sputtered Au film since the Au is deposited using the sputtering method. These factors can indeed cause errors in our FDTR measurements. For instance, the error of our measured interfacial thermal conductance (ITC) for fully activated samples is only ± 0.15 MW/m²K, which is very small compared to the corresponding ITC (i.e., 5.33 MW/m²K). This means the surface roughness, impurities, and crystal orientation of the sputtered Au film will slightly affect lattice vibrations in HKUST-1. However, the error of our measured ITC for saturated samples is large (i.e., ± 15.82 MW/m²K) compared to that of the corresponding ITC (i.e., 21.66 MW/m²K). It indicates that these factors mentioned above will largely affect the adsorption of water molecules in the interfacial region. Nevertheless, the mean ITC is found to increase from

5.33 MW/m²K to 21.66 MW/m²K when the samples adsorb saturated water molecules, implying that adsorbed water molecules can really affect the interfacial thermal transport between MOFs and substrates. In real applications, all the factors mentioned above are indeed unavoidable, while we think the mean ITC, which already includes all these factors, is suitable to characterize the thermal dissipation performance of the corresponding systems.

Based on the reviewer's comment, we have added more explanations in our revised manuscript:

- **(Page 6 in the revised manuscript):** This is reasonable, as it is verified by the error difference between the activated and saturated samples. The relative error of the measured ITC of the saturated samples is much larger than that of the activated samples, which is caused by the water adsorption in HKUST-1 when all the other measurement conditions are the same. Nevertheless, the mean ITC is found to increase from 5.33 MW/m²K to 21.66 MW/m²K when the samples adsorb saturated water, which also implies that the adsorbed water molecules can largely affect the interfacial thermal transport between MOFs and substrates.

Comment 3

HKUST-1 is a class of MOFs with large grain sizes, and the effectiveness of this method for MOFs with small grain sizes requires further consideration.

Our response:

We thank the reviewer for this good comment.

We agree that the grain size of HKUST-1 is large compared with many MOFs (e.g., MOF-801 [Science 356,430 (2017)], ZIF-8 [J. Am. Chem. Soc. 143, 12943 (2021)] and MOF-74 [Sci. Rep., **12**, 14735 (2022)] *et al.*). As mentioned above, the large grain size of HKUST-1 can ensure the accuracy of our FDTR measurements. Meanwhile, the laser spot radius for our FDTR apparatus is $\sim 3.6 \mu\text{m}$, which is smaller than the possible grain size of many MOFs. It is therefore possible to measure the thermal transport across small grain size MOFs and substrate using our FDTR apparatus. Furthermore, these MOFs with high water uptake capacity (e.g., MIL-101 [Nat. Commun. **11**, 2692 (2020)], MOF-303 [Nat. Protoc. **18**, 136 (2023)], MOF-808 [New J. Chem. **47**, 6433 (2023)], and UiO-66 [Angew.Chem. Int. Ed. **54**,11162 (2015)]) which are widely applied for passive cooling can be easily synthesized with large grain sizes from tens to hundreds of μm . At the same time, the morphology and grain size of MOFs may be controlled precisely by optimizing the corresponding synthesis procedure [Science (6149), 1230444. 341; Chem. Rev. **112**, 933 (2012)].

In this manuscript, we would like to systematically investigate the thermal transport across MOFs with/without water molecules and substrate, which is critical for the passive cooling applications as we discussed above (**Reviewer 2, Comment 0**). Our results show that the ITC across HKUST-1 and Au substrate can be improved largely. One of the main reasons for this is the introduced additional heat transfer channel between water molecules and the Au substrate. As many MOFs such as MIL-101, MOF-303, MOF-808, and UiO-66 used for passive cooling applications have high water uptake capacities, we believe our strategy here can also be applied to these MOFs. The results in this article may not only provide insights on the thermal transport across the MOFs with/without water molecules and substrate but also guide the design to improve the performance of the MOF-based cooling applications.

Based on the reviewer's comment, we have added more explanations in our revised manuscript:

- **(Page 3 in our revised manuscript):** Our findings provide new insights into thermal transport across MOFs (not only limited to HKUST-1) and their working objectives. We suggest a general strategy to introduce additional heat transfer channels between these MOFs and the objectives using adsorbed water, which will greatly benefit the performance of MOFs-related cooling applications.

REVIEWER COMMENTS

Reviewer #1 (Remarks to the Author):

I think the authors have addressed all the concerns from reviewers' comments. The manuscript is ready for publication. However, please check the following issues in the final version:

1. In the response, the authors indicate two continuous-wave (cw) lasers are used in the system. However, in the manuscript (page 4, line 87), an optical pump-probe spectroscopy based on the FDTR is used. Please check if the ultrafast expression is appropriate.
2. A new subsection S3.1 is added in the supplementary material. Please check the expression "the weighted average temperature distribution acquired by a probe beam at <frequency ω_1 > is".

Reviewer #2 (Remarks to the Author):

I acknowledge that exploiting interface resistances for tuning heat transfer is an effective strategy, where phonon wave interference can lead to an abnormal thermal conductivity behavior. However, the bottleneck for thermal transport in metal-organic frameworks is the interface between the nodes and the linkers, which leads to thermal transport dominated by Kapitza resistance. Since the authors repeatedly emphasize that the proposed strategy favors the application of MOF in cooling, it might be worthwhile for the authors to perform experimental confirmation to meet the expectations of the Nat. Commun. community. Considering that the simulation results are generally in good agreement with the experimental data, I suggest that the authors verify the generalizability of the proposed strategy on several additional MOFs.

Response Letter

Dear Reviewers,

Thank you for forwarding us the reviewers' reports on our manuscript entitled "Direct Observation of Tunable Thermal Conductance at Solid/porous Crystalline Solid Interfaces Induced by Water Adsorbents" (Manuscript ID: NCOMMS-23-10826). We appreciate your evaluation and believe that your comments have improved our revised manuscript substantially. In the following, we address all review comments (in cool blue for clarity) in detail and provide responses as well as the resulting revisions (in strong blue) to our manuscript. A copy of the manuscript with revisions highlighted in yellow is also provided for reference.

Response to the first referee

Comment 0

I think the authors have addressed all the concerns from reviewers' comments. The manuscript is ready for publication. However, please check the following issues in the final version.

Our response:

We thank the reviewer for the positive evaluation and recommendation for publication in Nature Communications. We have addressed all the comments point by point in the revised manuscript.

Comment 1

In the response, the authors indicate two continuous-wave (cw) lasers are used in the system. However, in the manuscript (page 4, line 87), an optical pump-probe spectroscopy based on the FDTR is used. Please check if the ultrafast expression is appropriate?

Our response:

We thank the reviewer for this comment.

We used "ultrafast" in the original manuscript to express that the frequency of the pump laser can be modulated to a high value. To avoid misunderstandings, we have deleted "ultrafast" in the revision.

Based on the reviewer's comment, we have updated the corresponding contents in the main text:

- **(Page 4 in main text):** We then characterized the thermal transport properties of Au/HKUST-1 heterointerfaces using an optical pump-probe spectroscopy based on the FDTR.

Comment 2

A new subsection S3.1 is added in the supplementary material. Please check the expression "the weighted average temperature distribution acquired by a probe beam at <frequency ω_1 > is"?

Our response:

We thank the reviewer for pointing this for us.

In the FDTR measurement, the probe beam detects the temperature distribution induced by the pump laser. Considering that the probe beam also has Gaussian intensity with a radius " ", the

measured temperature is the weighted average of the temperature on the surface [Rev. Sci. Instrum., **75**, 12 (2004)].

Based on the reviewer's comment, we have updated the corresponding contents in the main text:

- **(Page 9 in supplementary material):** Considering the probe laser intensity also has a Gaussian distribution, the weighted average temperature distribution acquired by a probe beam with a radius of r is

Response to the second referee

Comment 0

I acknowledge that exploiting interface resistances for tuning heat transfer is an effective strategy, where phonon wave interference can lead to an abnormal thermal conductivity behavior. However, the bottleneck for thermal transport in metal-organic frameworks is the interface between the nodes and the linkers, which leads to thermal transport dominated by Kapitza resistance. Since the authors repeatedly emphasize that the proposed strategy favors the application of MOF in cooling, it might be worthwhile for the authors to perform experimental confirmation to meet the expectations of the Nat. Commun. community.

Considering that the simulation results are generally in good agreement with the experimental data, I suggest that the authors verify the generalizability of the proposed strategy on several additional MOFs.

Our response:

We thank the reviewer for the very good comment.

As we mentioned in our main text and the previous response letter, the intensity radius for the probe and the pump laser beams are 5 μm and 3.6 μm , respectively. To ensure the accuracy of our FDTR measurements, the grain size of our synthesized MOF crystals should be larger than 10 μm . It has been shown that the UiO-66 [J. Am. Chem. Soc, **130**, 13850 (2008)] which is widely applied for water harvesting-based applications [J. Mater. Chem. A, **8**, 13364 (2020)] can be synthesized with large grain sizes from tens to hundreds of μm . We therefore choose UiO-66 as another example to carry out the measurements to demonstrate our strategy.

The UiO-66 crystals were prepared by modifying the method reported by Christopher A. Trickett [Angew.Chem. Int. Ed., **54**, 11162 (2015)]. In detail, a solvent of N, N-diethylformamide (DEF) was firstly dried by molecular sieves for several days. $\text{ZrOCl}_2 \cdot 8\text{H}_2\text{O}$ (0.037 mmol) and H_2BDC were separately dissolved in DEF (1 mL) by ultrasonication for 5 mins, respectively. They were then mixed in a glass vial which can be tightly capped, and the formic acid was added to the mixture to form a white color turbid solution. The solution was placed in the oven (preheated to 135 °C) for 48 hours. Block crystals were then obtained, followed by a washing process using DMF twice a day for two days. These bright crystals were activated under a vacuum at 60 °C for two days before use.

The SEM images (**Figure R1**) and the PXRD (**Figure R2**) show that the as-prepared UiO-66 possesses good crystallinity and its corresponding grain size ranges from several μm to hundreds of μm . It is noted that some peaks were not observed in the measured XRD spectrum compared to the simulated results. This is because the crystals are not ground into fine powders.

Figure R1 (Added as Figure S17). The SEM images of as-prepared UiO-66 crystals under different amplitudes, (a) 95× and (b) 1000×.

Figure R2 (Added as Figure S18). The PXR D patterns of UiO-66 crystals.

We then coated the Au film on the UiO-66 crystals as we did for HKUST-1 crystals and selected suitable samples for our FDTR measurements. Similarly, the saturated samples were prepared by immersing the activated UiO-66 crystals in water for ~ 2 hours. The sensitivity of FDTR measurements for Au/UiO-66 systems is quite similar to that of the Au/HKUST-1 systems because the crystalline UiO-66 has a similar thermal conductivity with the crystalline HKUST-1 (see the results below for details). Following the approach proposed by Babaei *et. al.* [Nat Commun, **11**, 4010 (2020)], the heat capacity of saturated UiO-66 is estimated basing $C_{saturated\ MOF} = C_{activated\ MOF} + \varphi C_{adsorbate}$, in which φ is the void fraction and has a value 47% for UiO-66 [ACS Appl. Mater. Interfaces, **11**, 38697 (2019)]. The calculated $C_{activated\ UiO-66}$ is 0.88 MJ/m³K based on MD simulations as suggested in Ref. [ACS Appl. Mater. Interfaces, **11**, 38697 (2019)]. The heat capacity of water C_{water} is 4.19 MJ/m³K which is directly taken from Ref. [Nat Commun, **11**, 4010 (2020)]. Therefore, the heat capacity of saturated UiO-66 can be calculated and has a value of ~ 2.9 MJ/m³K.

The thermal conductivity of UiO-66 and the ITC between UiO-66 and the Au film can be then obtained (**Figure R3**). Our results show that the thermal conductivity of UiO-66 is reduced from $0.67 \pm 0.13 \text{ W/mK}$ to $0.23 \pm 0.03 \text{ W/mK}$ when saturated water is adsorbed (**Figure R4**), and the corresponding ITC between UiO-66 crystals and the Au film increases from $16.4 \pm 2.9 \text{ MW/m}^2\text{K}$ to $19.7 \pm 3.15 \text{ MW/m}^2\text{K}$.

Figure R3 (Added as Figure S19). The thermal conductivity of UiO-66 at (a) activated and (b) saturated states. The interfacial thermal conductance (ITC) between UiO-66 and Au at (c) activated and (d) saturated states.

Figure R4 (Added as Figure S20). The thermal conductivity of UiO-66 and the interfacial thermal conductance (ITC) of UiO-66/Au samples with/without adsorbed water molecules.

Considering the time cost and synthesis challenges in experiments, we only measure one more MOF here. Meanwhile, molecular dynamics simulations can also be used to demonstrate our strategy as suggested by the reviewer: “Simulation results are generally in good agreement with the experimental data”.

We further performed nonequilibrium molecular dynamics (NEMD) simulations to calculate the interfacial thermal transport between MOF-505 and Au. The model used in NEMD simulations was built in the same way as we did for the Au/HKUST-1 systems, and the size of the system was 5.6 nm × 4.8 nm × 42.6 nm. A forcefield developed based on first-principles calculations was applied to describe the interatomic interactions of the MOF-505 framework [ACS Appl. Mater. Interfaces, **11**, 38697 (2019)]. The interaction among Au atoms and water molecules was the same as that for the Au/HKUST-1 systems with adsorbed water. The weak interaction parameters between Au and a water molecule in the **SI Note 4** were chosen to model the interaction between

Au and water. Other interaction parameters are the same as those used in the previous calculations of the Au/HKUST-1 heterointerface. The long-range electrostatic interactions were also considered. The NEMD simulation and water adsorption processes are the same as our previous calculations for Au/HKUST-1 systems with/without adsorbed water. Our results show that the ITC of activated Au/MOF-505 heterointerface is 7.72 ± 0.82 MW/m²K, and increases to 19.56 ± 1.81 MW/m²K when saturated water molecules are adsorbed (**Figure R5**).

In summary, the strategy utilizing adsorbed water we proposed here is a general way to enhance the interfacial thermal transport between MOF and substrates, e.g., the maximum enhancement is 7.1 times for Au/HKUST-1 interfaces, 1.7 times for Au/UiO-66 interfaces and 3.1 times for Au/MOF-505 interfaces. The enhancement depends on the morphologies and physical properties of MOFs and the corresponding interfaces.

Figure R5 (Added as Figure S21). The interfacial thermal conductance (ITC) of Au/MOF-505 heterointerfaces with/without adsorbed water molecules.

Based on the reviewer's comment, we have revised the main manuscript and supplementary information :

- **(Abstract in our revised manuscript):** Our experimental and/or modeling results show that the interfacial thermal conductance of Au/Cu₃(BTC)₂, Au/Zr₆O₄(OH)₄(BDC)₆ and Au/MOF-505 heterointerfaces is increased up to 7.1, 1.7 and 3.1 folds by this strategy, respectively, where Cu₃(BTC)₂ is referred to as HKUST-1 and Zr₆O₄(OH)₄(BDC)₆ is referred to as UiO-66.
- **(Page 3 in our revised manuscript):** Our frequency-domain thermoreflectance (FDTR)^{32,33} measurements and/or molecular dynamics (MD) simulations show that the ITC of Au/HKUST-1, Au/UiO-66 and Au/MOF-505 heterointerfaces can be improved from 5.3 MW/m²K, 12.5 MW/m²K and 6.9 MW/m²K to 37.5 MW/m²K, 22.9 MW/m²K and 21.4 MW/m²K (~7.1, ~1.7 and ~3.1 times) via this strategy, respectively.
- **(Page 9 in our revised manuscript):** To show the generalizability of our proposed strategy, we further investigated the thermal transport in two more MOF/substrate systems (i.e., UiO-66/Au and MOF-55/Au heterointerfaces) using either experiments or MD simulations. The chosen UiO-66^{44,45} and MOF-505⁴⁶ here show good water adsorption capacity and stability. Our FDTR measurements show that the ITC between UiO-66 crystals and the Au film increases from 16.4±2.9 MW/m²K to 19.7±3.15 MW/m²K when saturated water is adsorbed (see S5.1 in SI Note 5 for details). Our MD simulations also show that the ITC of activated Au/MOF-505 heterointerface is 7.72±0.82 MW/m²K, and increases to

19.56±1.81 MW/m²K when saturated water molecules are adsorbed (see S5.2 in SI Note 5 for details).

- **(Conclusions in our revised manuscript):** In summary, we have designed a tunable strategy utilizing the adsorbed water in porous MOFs to manipulate the thermal transport across Au/MOF interfaces. Our FDTR measurements and/or simulations showed that a maximum ITC ~37.9 MW/m²K, ~22.9 MW/m²K, and ~21.4 MW/m²K could be achieved when saturated water molecules were adsorbed in HKUST-1, UiO-66, and MOF-505, respectively. These values were ~7.1, ~1.7, and ~3.1 times higher than the ITCs of the activated Au/HKUST-1, Au/UiO-66, and Au/MOF-505 heterointerfaces.
- **(Add one more section Supplementary Note 5 in the revised supplementary information):**

Supplementary Note 5. The generalizability of the proposed strategy

To show the generalizability of our proposed strategy, we further investigated the thermal transport in two other MOF/substrate systems (i.e., UiO-66/Au and MOF-55/Au interfaces) using either experiments or MD simulations. The chosen UiO-66^{21,22} and MOF-505²³ here show good water adsorption capacity and stability.

S 5.1 The synthesis of Au/UiO-66 samples and the corresponding FDTR measurement

The UiO-66 crystals were prepared by modifying the method reported by Christopher A. Trickett²⁴. In detail, a solvent of N, N-diethylformamide (DEF) was firstly dried by molecular sieves for several days. Then, ZrOCl₂·8H₂O (0.037 mmol) and H₂BDC were dissolved separately in DEF (1 mL) by ultrasonication for 5 mins. They were then mixed in a glass vial, which can be tightly capped, and the formic acid was added to the mixture

to form a white color turbid solution. The solution was placed in the oven (preheated to 135 °C) for 48 hours. Block crystals were then obtained, followed by a washing process using DMF twice a day for two days. These bright crystals were activated under a vacuum at 60 °C for two days before use. The SEM images (**Figure S17**) and the PXRD (**Figure S18**) show that the as-prepared UiO-66 possesses good crystallinity, with grain size ranging from several μm to hundreds of μm . It is noted that some peaks were not observed in the measured XRD spectrum compared to the simulated results. This is because the crystals are not ground into fine powders.

We then coated the Au film on the UiO-66 crystals as we did for HKUST-1 crystals and selected suitable samples for our FDTR measurements. Similarly, the saturated samples were prepared by immersing the activated UiO-66 crystals in water for ~2 hours. The sensitivity of FDTR measurements for Au/UiO-66 systems is quite similar to that of the Au/HKUSTS-1 systems because the crystalline UiO-66 has a similar thermal conductivity with the crystalline HKUST-1 (see the results below for details). Following the approach proposed by Babaei *et. al.*¹, the heat capacity of saturated UiO-66 is estimated basing $C_{\text{saturated MOF}} = C_{\text{activated MOF}} + \varphi C_{\text{adsorbate}}$, in which φ is the void fraction and has a value of 47% for UiO-66²². The calculated $C_{\text{activated UiO-66}}$ is 0.88 MJ/m³K based on MD simulations as suggested in Ref.²². The heat capacity of water C_{water} is 4.19 MJ/m³K which is directly taken from Ref.¹. Therefore, the heat capacity of saturated UiO-66 can be calculated and has a value of ~2.9 MJ/m³K. The thermal conductivity of UiO-66 and the ITC between UiO-66 and the Au film can be then obtained (**Figure S19**). Our results show that the thermal conductivity of UiO-66 is reduced from 0.67±0.13 W/mK to 0.23±0.03 W/mK

when saturated water is adsorbed (**Figure S20**), and the corresponding ITC between UiO-66 crystals and the Au film increases from 16.4 ± 2.9 MW/m²K to 19.7 ± 3.15 MW/m²K.

S 5.2 Thermal transport across Au/MOF-505 interfaces

To further verify the enhancement of interfacial thermal transport on other MOFs constructed heterointerfaces, we further performed nonequilibrium molecular dynamics (NEMD) simulations to calculate the interfacial thermal transport between MOF-505 and Au. The model used in NEMD simulations was built in the same way as we did for the Au/HKUST-1 systems, and the size of the system was $5.6 \text{ nm} \times 4.8 \text{ nm} \times 42.6 \text{ nm}$. A forcefield developed based on first-principles calculations was applied to describe the interatomic interactions of the MOF-505 framework²². The interaction among Au atoms and water molecules was the same for the Au/HKUST-1 systems with adsorbed water. The weak interaction parameters between Au and a water molecule in the **SI Note 4** were chosen to model the interaction between Au and water. Other interaction parameters are the same as those used in the previous calculations of the Au/HKUST-1 heterointerface. The long-range electrostatic interactions were also considered. The NEMD simulation and water adsorption processes are the same as our previous calculations for Au/HKUST-1 systems with/without adsorbed water. Our results show that the ITC of activated Au/MOF-505 heterointerface is 7.72 ± 0.82 MW/m²K, and increases to 19.56 ± 1.81 MW/m²K when saturated water molecules are adsorbed (**Figure S21**).

In summary, the strategy utilizing adsorbed water we proposed here is a general way to enhance the interfacial thermal transport between MOF and substrates, e.g., the maximum enhancement is 7.1 times for Au/HKUST-1 interfaces, 1.7 times for Au/UiO-66 interfaces

and 3.1 times for Au/MOF-505 interfaces. The enhancement depends on the morphologies and physical properties of MOFs and the corresponding interfaces.

[1] Babaei, H. *et al.* Observation of reduced thermal conductivity in a metal-organic framework due to the presence of adsorbates. *Nat. Commun.* **11**, 4010 (2020).

[21] Cavka, J. H. *et al.* A new zirconium inorganic building brick forming metal organic frameworks with exceptional stability. *J. Am. Chem. Soc.* **130**, 13850–13851 (2008).

[22] Wieme, J. *et al.* Thermal engineering of metal–organic frameworks for adsorption applications: a molecular simulation perspective. *ACS Appl. Mater. Interfaces* **11**, 38697–38707 (2019).

[23] Qi, Z.-P., Yang, J.-M., Kang, Y.-S., Guo, F. & Sun, W.-Y. Facile water-stability evaluation of metal–organic frameworks and the property of selective removal of dyes from aqueous solution. *Dalton Trans.* **45**, 8753–8759 (2016).

[24] Trickett, C. A. *et al.* Definitive molecular level characterization of defects in UiO-66 crystals. *Angew. Chem. Int. Ed.* **54**, 11162–11167 (2015).

REVIEWER COMMENTS

Reviewer #2 (Remarks to the Author):

Even though the authors verified that the interfacial heat transfer was enhanced for other MOFs, the much larger thermal resistance resulted by the low thermal conductivity of MOFs (just as the Figure R4 suggested) would suppress the effect manifestation of increasing interfacial thermal conductance on MOF-related applications such as cooling. Therefore, we suggest that the authors conduct experiments on applying MOFs for cooling to demonstrate the effectiveness of the proposed strategy.

Response Letter

Dear Reviewers,

Thank you for forwarding us the reviewers' reports on our manuscript entitled "Direct Observation of Tunable Thermal Conductance at Solid/porous Crystalline Solid Interfaces Induced by Water Adsorbents" (Manuscript ID: NCOMMS-23-10826). We appreciate your evaluation and believe that your comments have improved our revised manuscript substantially. In the following, we address all review comments (in cool blue for clarity) in detail and provide responses as well as the resulting revisions (in strong blue) to our manuscript. A copy of the manuscript with revisions highlighted in yellow is also provided for reference.

Response to the second referee

Comment 0

Even though the authors verified that the interfacial heat transfer was enhanced for other MOFs, the much larger thermal resistance resulted by the low thermal conductivity of MOFs (just as the Figure R4 suggested) would suppress the effect manifestation of increasing interfacial thermal conductance on MOF-related applications such as cooling. Therefore, we suggest that the authors conduct experiments on applying MOFs for cooling to demonstrate the effectiveness of the proposed strategy.

Our response:

We thank the reviewer for this comment.

It is known that the interfacial thermal transfer is critical for the heat dissipation of these devices with interfaces [Rev. Mod. Phys. **94**, 025002 (2022); Nat. Commun., **5**, 3435 (2014); Nat. Commun., **13**, 4901 (2022); Nat. Commun., **12**, 1284 (2021); Nat. Mater., **11**, 502-506 (2012); Sci. Adv., **7**, eabf8197 (2021); Adv. Mater. **26**, 6093–6099 (2014)]. For example, a previous study showed that the maximum temperature of β -Ga₂O₃/substrate heterostructures decreased from 1860 K to 1230 K when the interfacial thermal conductance (ITC) of β -Ga₂O₃/substrate heterostructures increased from ~10 MW/m²K to ~50 MW/m²K [IEEE Trans. Compon. Packag. Manuf. **12**, 638 (2022)]. In this manuscript, we propose this strategy to tune the interfacial thermal transport between the substrate and MOFs, which should benefit their potential cooling applications. We here focus on the underlying mechanisms that cause the enhancement of ITC between the substrate and MOFs and propose a possible strategy based on these fundamental understandings to tune the corresponding ITC. We hope the strategy proposed here can motivate the researchers in this area to design and apply real cooling applications based on our strategy.

In MOF-related cooling applications, the substrate (e.g., solar panel or electronics) and MOF are treated as heat sources and a heat spreader, respectively. Therefore, the heat transfer from the heat source to the ambient air can be divided into three steps (details can be found in **rebuttal letter 1**): 1) The heat is generated on the Au surface and conducted within the Au substrate; 2) The heat conductively transfers across the interface from the Au substrate to the HKUST-1; 3) The heat adsorbed in HKUST-1 will dissipate the ambient air mainly through the evaporation of the adsorbed water owing to its high enthalpy. As we mentioned in the introduction of our main text, MOFs typically possess a low thermal conductivity below 2 W/mK at room temperature [Dalton Trans. **46**, 13342–13344 (2017); Adv. Mater. **27**, 3453–3459 (2015) and Int. J. Heat Mass Transf. **50**, 405–411 (2007)] (e.g., ~0.05 W/mK for MIL-101 (Cr) [Microporous and Mesoporous Mater.,

244, 180–191 (2017)] and ~ 0.04 W/mK for MOF-801 [Sustain. Mater. Techno. **32**, e00442 (2022)]), and are therefore regarded as poor thermal conductors. This is also demonstrated in our FDTR measurements, the thermal conductivity at room temperature is ~ 0.742 W/mK for HKUST-1 and ~ 0.67 W/mK for UiO-66, respectively. Therefore, in step 3, the heat dissipated through the conduction heat transfer is small compared to that carried by the evaporative heat transfer due to the very low thermal conductivity of MOFs with and without water as we discussed above. For instance, even though the room temperature thermal conductivity of MIL-101 (Cr) is only 0.05 W/mK, the cooling performance of MIL-101 (Cr) based coating can be as high as ~ 10 °C owing to its high equivalent enthalpy (i.e., 1950 J/g_{coating}) [Joule **4**, 435 (2020)].

As the heat must be transferred from the substrate to MOFs in the above cooling applications, the thermal transport across the MOF with or without atmospheric water/substrate (i.e., solar panel or electronics) is critical for the corresponding cooling performance in these applications mentioned above. For instance, the heat current transferred from the substrate (i.e., the heat source) to the MOF (i.e., can be regarded as the heat dissipator) will be increased seven times if the ITC between the substrate and the MOF improves seven times as $Q = G \times \Delta T$. We, therefore, believe that this strategy proposed in the manuscript to tune the interfacial thermal transfer between substrate and MOFs can benefit these MOF-based cooling applications a lot.

Furthermore, our previous study applied a MOF-801-based cooling coating to cool the solar panels [Droplet **2**, e32 (2023)]. In that application, we paint the MOF-801-based cooling coating on the backside of the solar panels and utilize the atmospheric water adsorption-desorption process in the coating to cool the temperature of the solar panels. It has been demonstrated that MOF-801 shows an excellent water uptake capacity at low relative humidity (RH) [Science **434**, 430 (2017)], which indicates a good cooling performance at low RH of the MOF-801-based coatings. Our results show

that the temperature of solar panels can be reduced by over 10 °C at most at a RH of 28% using such a strategy. Meanwhile, Wang *et al.* [Joule 4, 435 (2020)] also utilize the atmospheric water adsorption and desorption process of MIL-101(Cr) for chips' cooling and find that the maximum cooling temperature can be ~10 °C.

Based on the reviewer's comment, we have added more explanations in the main text:

- **(Page 1 in Abstract):** Improving interfacial thermal transport is crucial for heat dissipation in devices with interfaces, such as electronics, buildings, and solar panels. Here, we design a strategy by utilizing the water adsorption-desorption process in porous metal-organic frameworks (MOFs) to tune the interfacial heat transfer, which could benefit their potential in cooling or heat dissipation applications.
- **(Pages 1-2 in Abstract):** Our findings revealed the underlying mechanisms for tailoring thermal transport at the solid/porous MOF heterointerfaces by water adsorbents, which could motivate and benefit the new cooling system design based on MOFs.
- **(Page 11 in CONCLUSIONS):** The underlying mechanism on the heat transfer across the MOFs/solid interfaces provided here will guide the design of effective cooling or heat dissipation systems using MOFs.

REVIEWERS' COMMENTS

Reviewer #2 (Remarks to the Author):

I think the authors have addressed all the concerns from reviewers' comments. The manuscript is ready for publication. In MOF-related cooling applications, part of the heat generated by the heat source is first transferred to MOFs and then transferred to the water to drive the water evaporation. Therefore, the heat conductivity affects heat transfer to some extent. But the proposed strategy can enable the adsorbed water gather at the interfacial region, causing that more heat can be directly transferred to the water for evaporation and the heat transfer rate can indeed be accelerated. It is worth noting that the premise of the statement "the heat current transferred from the substrate (i.e., the heat source) to the MOF (i.e., can be regarded as the heat dissipator) will be increased seven times if the ITC between the substrate and the MOF improves seven times as $Q = G \times \Delta T$." is that the temperature difference does not change.

Response Letter

Dear Reviewers,

Thank you for forwarding us the reviewers' reports on our manuscript entitled "Direct Observation of Tunable Thermal Conductance at Solid/porous Crystalline Solid Interfaces Induced by Water Adsorbents" (Manuscript ID: NCOMMS-23-10826). We appreciate your evaluation and believe that your comments have improved our revised manuscript substantially. In the following, we address all review comments (in cool blue for clarity) in detail and provide responses as well as the resulting revisions (in strong blue) to our manuscript. A copy of the manuscript with revisions highlighted in yellow is also provided for reference.

Response to the second referee

Comment 0

I think the authors have addressed all the concerns from reviewers' comments. The manuscript is ready for publication. In MOF-related cooling applications, part of the heat generated by the heat source is first transferred to MOFs and then transferred to the water to drive the water evaporation. Therefore, the heat conductivity affects heat transfer to some extent. But the proposed strategy can enable the adsorbed water gather at the interfacial region, causing that more heat can be directly transferred to the water for evaporation and the heat transfer rate can indeed be accelerated. It is worth noting that the premise of the statement "the heat current transferred from the substrate (i.e., the heat source) to the MOF (i.e., can be regarded as the heat dissipator) will be increased seven

times if the ITC between the substrate and the MOF improves seven times as $Q = G \times \Delta T$." is that the temperature difference does not change.

Our response:

We thank the reviewer for this comment.

We agree that when the ITC between the substrate and MOF is increased seven times of which the premise is that the interface temperature difference remains unchanged.